# Deep learning to estimate lung disease mortality from chest radiographs

Jakob Weiss[1,2,3,4,8], Vineet K. Raghu [1,4,8], Dennis Bontempi[1,5], David C. Christiani [6,7], Raymond H. Mak [1,2], Michael T. Lu [1,4,9] & Hugo J.W.L. Aerts [1,2,4,7,9] ✉

Prevention and management of chronic lung diseases (asthma, lung cancer, etc.) are of great importance. While tests are available for reliable diagnosis, accurate identification of those who will develop severe morbidity/mortality is currently limited. Here, we developed a deep learning model, CXR Lung-Risk, to predict the risk of lung disease mortality from a chest x-ray. The model was trained using 147,497 x-ray images of 40,643 individuals and tested in three independent cohorts comprising 15,976 individuals. We found that CXR Lung-Risk showed a graded association with lung disease mortality after adjustment for risk factors, including age, smoking, and radiologic findings (Hazard ratios up to 11.86 [8.64–16.27]; p < 0.001). Adding CXR Lung-Risk to a multivariable model improved estimates of lung disease mortality in all cohorts. Our results demonstrate that deep learning can identify individuals at risk of lung disease mortality on easily obtainable x-rays, which may improve personalized prevention and treatment strategies.

Prevention and management of chronic lung diseases such as COPD, asthma, or lung cancer are of great importance given their high prevalence and the associated economic burden on the healthcare system[1–5]. While dedicated tests are available for reliable diagnosis and monitoring of lung diseases[6–8], accurate prediction to identify those who will eventually develop severe morbidity and mortality is currently limited. Therefore, new methods to improve risk stratification are desirable. Chest radiographs (CXR) are the most common diagnostic imaging test and are acquired in the workup of many lung diseases[9]. However, although most of them are without actionable radiological findings by a human reader[10], especially in the early stages of lung disease, more quantitative, computer-aided analyses may provide a

window into the risk and extent of lung disease beyond established methods.

With recent advances in artificial intelligence, new possibilities to automatically capture and quantify a multitude of information have become available[11,12]. This is particularly true in medical imaging, where deep learning (convolutional neural networks or CNNs) has demonstrated high performance in estimating the risk of mortality, incident lung cancer, or biological aging from a chest radiograph image[13–15]. These results indicate that medical imaging might be helpful to personalize risk assessment based on changes to our anatomy, even in an asymptomatic preclinical stage[16–19]. Moreover, using medical imaging for risk estimation may have broader applications compared to established methods in clinical care as imaging-based measures can be

[1]Artificial Intelligence in Medicine (AIM) Program, Mass General Brigham, Harvard Medical School, Harvard Institutes of Medicine, 77 Avenue Louis Pasteur, Boston, MA 02115, USA. [2]Department of Radiation Oncology, Brigham and Women's Hospital, Dana-Farber Cancer Institute, Harvard Medical School, 75 Francis Street and 450 Brookline Avenue, Boston, MA 02115, USA. [3]Department of Diagnostic and Interventional Radiology, University Medical Center Freiburg, Faculty of Medicine, University of Freiburg, Hugstetter Str. 55, 79106 Freiburg, Germany. [4]Cardiovascular Imaging Research Center, Massachusetts General Hospital, Harvard Medical School, 165 Cambridge Street, 02114 Boston, USA. [5]Radiology and Nuclear Medicine, CARIM & GROW, Maastricht University, Universiteitssingel 40, 6229 ER Maastricht, The Netherlands. [6]Department of Environmental Health, Harvard T.H. Chan School of Public Health, 655 Huntington Ave., Boston, MA 02115, USA. [7]Pulmonary and Critical Care Division, Massachusetts General Hospital, Harvard Medical School, 55 Fruit Street, Boston, MA 02114, USA. [8]These authors contributed equally: Jakob Weiss, Vineet K. Raghu. [9]These authors jointly supervised this work: Michael T. Lu, Hugo J.W.L. Aerts. ✉e-mail: Hugo_Aerts@DFCI.harvard.edu

calculated opportunistically from existing scans acquired in daily routine[13,20–22].

In this study, we developed a CNN (CXR-Lung-Risk) to identify individuals at high risk for lung disease mortality. The only input to the model is a single existing chest radiograph and the output is a risk for lung disease mortality expressed in years meaning, if the model outputs a risk of 75 years this is an equal risk of lung disease mortality as the risk of an average 75-year-old individual. We tested the prognostic value of CXR-Lung-Risk in three distinct clinical scenarios, including an asymptomatic community population enrolled in the Prostate, Lung, Colorectal, Ovarian (PLCO) Cancer Screening Trial[23,24], heavy smokers eligible for lung cancer screening CT enrolled in the National Lung Screening Trial (NLST)[25] and patients with histologically confirmed early-stage (I-III) lung cancer from the Boston Lung Cancer Study (BLCS). Our findings motivate the use of deep learning to identify individuals at high risk of lung disease mortality from easily obtainable and low-cost chest radiograph images. These findings may allow for improved risk assessment of those who would benefit most from personalized prevention and treatment strategies.

## Results

We developed a deep learning model to estimate the risk of lung disease mortality using a chest radiograph as the only input and independently tested the model in three held-out datasets comprising more than 15,000 individuals: I) 20% of participants (n = 10,155, median follow-up=17.0 [IQR 14.8–19.0] years) not seen during model development from PLCO[23,24]. PLCO was a multicenter randomized controlled trial of chest radiography for cancer screening in asymptomatic individuals aged 55–74 years enrolled at 10 US sites from 1993 through 2001. Outcomes were assessed via annual questionnaires, communication with next of kin and the National Death Index. Cause of death was determined using ICD-9 codes. II) Participants from the NLST[25] chest radiograph arm (n = 5,414; median follow-up=11.9 [IQR 7.3–12.3] years). NLST was a randomized controlled trial that enrolled heavy

smokers (≥30 pack years) aged 55–74 years for lung cancer screening via chest CT vs. chest radiograph at 21 US sites from 2002 through 2004. Similar to PLCO, outcomes were assessed via annual questionnaires, communication with next of kin and the National Death Index and ICD 9 codes were used to determine cause of death. III) patients from the BLCS (n = 407; median follow-up=3.4 [IQR 1.5–7.2] years), which is an ongoing multicenter observational epidemiologic cohort registry of patients with histologically confirmed lung cancer. Mortality was verified by study staff via manual chart review and was available for lung cancer-specific mortality only. An overview of the study design and analyses is provided in Fig. 1.

PLCO had the lowest mean CXR-Lung-Risk (mean 63.0 ± 5.5 years), followed by NLST (screening eligible heavy smokers; [mean 66.1 ± 5.7 years]) and then BLCS (patients with histologically confirmed lung cancer [mean 70.5 ± 6.7]) (p < 0.001 for all comparisons). In general, CXR-Lung-Risk was significantly higher in men, current or former smokers, and if traditional radiographic findings were present (Supplementary Figs. 1 and 2). Further detailed patient demographics for all datasets are provided in Tables 1a, b and 2, and Supplementary Table 1.

Internal testing in PLCO: First, CXR-Lung-Risk was independently tested in the remaining held-out dataset of PLCO (n = 10,155) not seen during any part of training. Kaplan-Meier survival analysis revealed a graded association between CXR-Lung-Risk categories and lung disease mortality (Fig. 2a). The univariable hazard ratio for lung disease mortality for those with a CXR-Lung-Risk between 65 and 75 years old was 5.74 [4.69–7.01]; p < 0.001 and 31.45 [24.43–40.48]; p < 0.001 for those >75 years old compared to the reference group (CXR-Lung-Risk <65 years old). This association remained robust after adjusting for baseline demographics (chronological age, sex, race, smoking status, pack years, body mass index) and clinical risk factors (prevalent diabetes mellitus, hypertension, history of stroke, myocardial infarction and cancer) (adjusted hazard ratio for CXR-Lung-Risk between 65–75 years old was 3.52 [2.81–4.41]; p < 0.001 and 11.86 [8.64–16.27]; p < 0.001 for those >75-year-old).

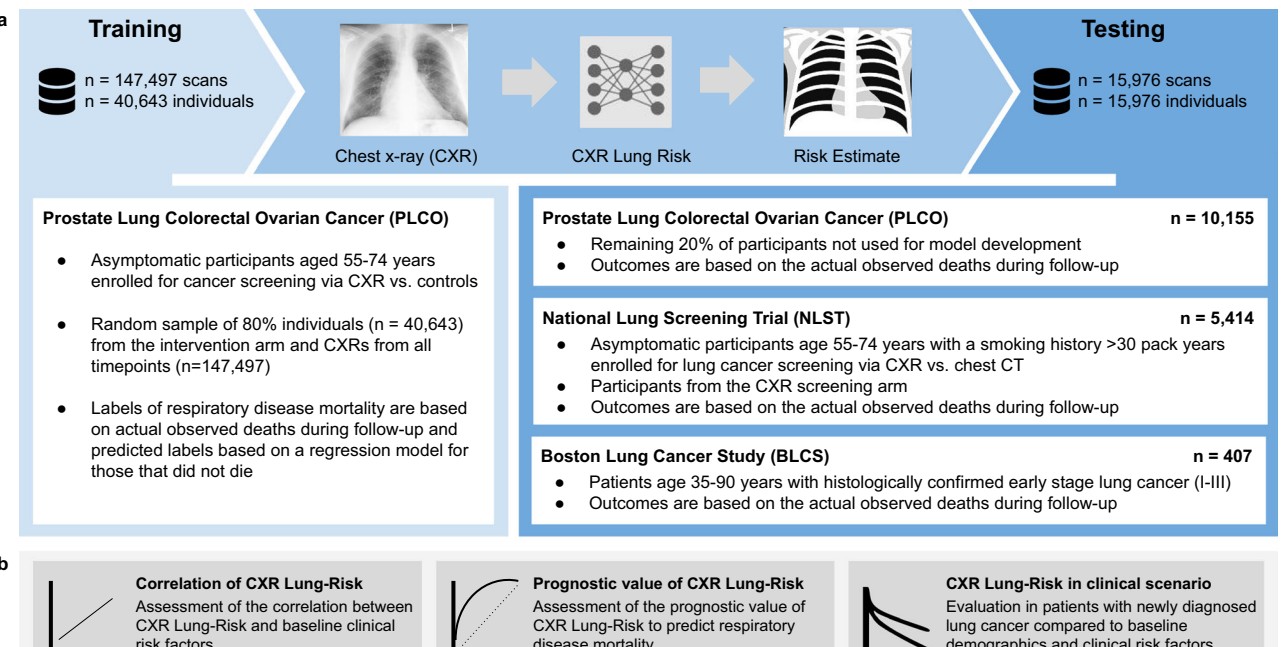

**Fig. 1 | Overview of the study design. a** The CXR-Lung-Risk model was developed in PLCO. The only input to the model is a chest radiograph image; the model output is an estimated risk of lung disease mortality. Independent testing was performed in a held-out subset of PLCO participants, individuals enrolled in NLST and patients with histologically confirmed lung cancer from the BLCS. **b** The prognostic performance of the CXR-Lung-Risk model was evaluated and compared to clinical risk factors in all datasets. Source data are provided as a Source Data file. PLCO Prostate, Lung, Colorectal, Ovarian Cancer Screening Trial; NLST National Lung Screening Trial, BLCS Boston Lung Cancer Study.

**Table 1 | (a) Patient demographics and clinical risk factors of PLCO and NLST participants. (b) Radiology findings of PLCO and NLST participants**

| Variables | PLCO testing dataset | | | | | NLST | | | | |
|---|---|---|---|---|---|---|---|---|---|---|
| | Entire dataset | Lung-Risk < 65 years | Lung-Risk ≥ 65-<75 years | Lung-Risk > 75 years | p | Entire dataset | Lung-Risk < 65 years | Lung-Risk ≥65-<75 years | Lung-Risk > 75 years | p |
| **(a)** | | | | | | | | | | |
| N | 10,155 | 7182 | 2667 | 306 | | 5414 | 2600 | 2433 | 381 | |
| Age (years) | 62.4±5.4 | 60.9±4.8 | 65.7±5.1 | 67.0±4.6 | <0.001 | 61.7±5.0 | 59.6±4.0 | 63.3±5.0 | 66.1±5.1 | <0.001 |
| CXR-Lung-Risk | 63.0±5.5 | 60.3±3.0 | 68.2±2.5 | 80.3±5.1 | <0.001 | 66.1±5.7 | 61.6±2.4 | 68.8±2.6 | 79.4±3.8 | <0.001 |
| Sex (m) | 51.6% (5232/10,155) | 45.5% (3271/7182) | 64.6% (1724/2667) | 77.5% (237/306) | <0.001 | 55.4% (2998/5414) | 45.2% (1175/2600) | 63.4% (1542/2433) | 73.8% (281/381) | <0.001 |
| Race | | | | | | | | | | |
| Non-Hispanic White | 86.8% (8810/10,155) | 86.9% (6243/7182) | 85.6% (2284/2667) | 92.5% (283/306) | 0.03 | 93.1% (5040/5414) | 93% (2418/2600) | 93.1% (2264/2433) | 94.0% (358/381) | 0.97 |
| Non-Hispanic Black | 6.0% (611/10,155) | 5.8% (420/7182) | 6.6% (177/2667) | 4.6% (14/306) | | 3.8% (207/5414) | 3.9% (101/2600) | 3.9% (94/2433) | 3.1% (12/381) | |
| Hispanic | 2.0% (202/10,155) | 1.9% (139/7182) | 2.2% (60/2667) | 1.0% (3/306) | | NA | NA | NA | NA | |
| Other | 5.2% (532/10,155) | 5.3% (380/7182) | 5.5% (146/2667) | 1.9% (6/306) | | 3.1% (167/5414) | 3.1% (81/2600) | 3.1% (75/2433) | 2.9% (11/381) | |
| Obesity (BMI > 30 kg/m2) | 24.2% (2460/10,155) | 25.2% (1809/7182) | 22.4% (598/2667) | 17.3% (53/306) | <0.001 | 27.6% (1493/5414) | 28.3% (735/2600) | 28.8% (700/2433) | 15.2% (58/381) | <0.001 |
| Smoking | | | | | | | | | | |
| Never | 45.8% (4650/10,155) | 51.9% (3727/7182) | 33.5% (895/2667) | 9.2% (28/306) | <0.001 | NA | NA | NA | NA | NA |
| Former | 43.2% (4391/10,155) | 40.9% (2939/7182) | 48.3% (1287/2667) | 53.9% (165/306) | | 50.4% (2728) | 54.0% (1403/2600) | 48.3% (1175/2433) | 39.4% (150/381) | <0.001 |
| Current | 11.0% (1114/10,155) | 7.2% (516/7182) | 18.2% (465/2667) | 36.9% (113/306) | | 49.6% (2686/5414) | 46.0% (1197/2600) | 51.7% (1258/2433) | 60.6% (231/381) | |
| Pack years | 19.2±27.7 | 14.1±22.6 | 28.9±32.5 | 55.9±39.7 | <0.001 | 55.8±23.5 | 51.4±21.1 | 58.6±24.2 | 67.2±27.4 | <0.001 |
| Diabetes | 7.1% (718/10,155) | 6.3% (456/7182) | 8.9% (238/2667) | 7.8% (24/306) | <0.001 | 9.3% (501/5414) | 9.0% (234/2600) | 9.7% (235/2433) | 8.4% (32/381) | 0.58 |
| Hypertension | 32.8% (3333/10,155) | 30.9% (2217/7182) | 37.4% (997/2667) | 38.9% (119/306) | <0.001 | 36.8% (1985/5414) | 32.7% (849/2600) | 40.9% (991/2433) | 38.2% (145/381) | <0.001 |
| Past Myocardial infarction | 8.8% (893/10,155) | 5.9% (421/7182) | 15.5% (413/2667) | 19.3% (59/306) | <0.001 | 12.3% (665/5414) | 8.4% 8218/2600) | 15.3% (370/2433) | 20.3% (77/381) | <0.001 |
| Past Stroke | 2.4% (246/10,155) | 1.8% (127/7182) | 3.8% (102/2667) | 5.6% (17/306) | <0.001 | 3.2% (173/5414) | 2.4% (61/2600) | 3.7% (89/2433) | 6.1% (23/381) | <0.001 |
| Past Cancer | 4.1% (414/10,155) | 3.9% (278/7182) | 4.8% (127/2667) | 2.9% (9/306) | 0.08 | 4.2% (225/5414) | 4.1% (106/2600) | 4.2% (103/2433) | 4.2% (16/381) | 0.96 |
| Lung disease mortality | 5.2% (523/10,155) | 2.0% (147/7182) | 10.2% (271/2667) | 34.3% (105/306) | <0.001 | 7.0% (378/5414) | 3.0% (79/2600) | 8.5% (208/2433) | 23.9% (91/381) | <0.001 |
| Lung cancer-related mortality | 2.5% (252/10,155) | 1.0 % (70/7182) | 5.1% (135/2667) | 15.4% (47/306) | <0.001 | 5.0% (270/5414) | 2.7% (69/2600) | 6.3% (154/2433) | 12.3% (47/381) | <0.001 |
| Follow-up in years (IQR) | 17.0 (14.8-19.0) | 17.2 (15.5-19.1) | 16.3 (12.4-18.7) | 11.2 (6.2-16.0) | <0.001 | 11.9 (7.6-12.3) | 12.0 (11.2-12.4) | 11.9 (7.3-12.3) | 8.8 (6.1-12.0) | <0.001 |
| **(b)** | | | | | | | | | | |
| N | 10,155 | 7182 | 2667 | 306 | | 5414 | 2600 | 2433 | 381 | |
| Nodule | 16.6% (1684/10,155) | 14.5% (1039/7182) | 21.1% (563/2667) | 26.8% (82/306) | <0.001 | 20.4% (1104/5414) | 17.0% (441/2600) | 22.4% (546/2433) | 30.7% (117/381) | <0.001 |
| Atelectasis | 0.2% (18/10,155) | 0.08% (6/7182) | 0.4% (11/2667) | 0.3% (1/306) | 0.002 | 0.3% (16/5414) | 0.1% (3/2600) | 0.5% (11/2433) | 0.5% (2/381) | 0.06 |
| Pleural fibrosis | 8.0% (811/10,155) | 5.7% (410/7182) | 12.5% (334/2667) | 21.9% (67/306) | <0.001 | 4.9% (266/5414) | 3.5% (90/2600) | 6.1% (149/2433) | 7.1% (27/381) | <0.001 |
| Lung fibrosis | 17.2% (1746/10,155) | 12.8% (919/7182) | 25.9% (691/2667) | 44.4% (136/306) | <0.001 | 6.8% (370/5414) | 4.2% (110/2600) | 8.0% (194/2433) | 17.3% (66/381) | <0.001 |
| COPD/Emphysema | 6.3% (639/10,155) | 3.4% (246/7182) | 10.7% (285/2667) | 35.3% (108/306) | <0.001 | 14.9% (809/5414) | 8.8% (228/2600) | 17.3% (420/2433) | 42.3% (161/381) | <0.001 |
| Lung Opacity | 2.6% (259/10,155) | 1.8% (126/7182) | 3.9% (105/2667) | 9.2% (28/306) | <0.001 | 0.2% (9/5414) | 0.04% (1/2600) | 0.2% (6/2433) | 0.5% (2/381) | 0.04 |
| Cardiac Abnormality | 9.4% (950/10,155) | 6.8% (486/7182) | 15.9% (423/2667) | 13.4% (41/306) | <0.001 | 1.1% (59/5414) | 0.5% (13/2600) | 1.4% (34/2433) | 3.1% (12/381) | <0.001 |
| Lymphadenopathy | 1.2% (121/10,155) | 0.8% (61/7182) | 2.0% (53/2667) | 2.3% (7/306) | <0.001 | 0.3% (16/5414) | 0.2% (5/2600) | 0.4% (9/2433) | 0.5% (2/381) | 0.35 |
| Bone/Chest Wall Lesion | 9.8% (995/10,155) | 8.7% (623/7182) | 12.4% (332/2667) | 13.1% (40/306) | <0.001 | 0.4% (22/5414) | 0.2% (5/2600) | 0.6% (14/2433) | 0.8% (3/381) | 0.05 |

PLCO Prostate, Lung, Colorectal, Ovarian Cancer Screening Trial, NLST National Lung Screening Trial, IQR interquartile range.
Table 1a: Patient demographics and clinical risk factors of PLCO and NLST patients for the entire data set and stratified by CXR-Lung-Risk groups. As appropriate, the chi-square test, student's t-test, Wilcoxon test or Kruskal Wallis test were calculated. All p values are two-sided.
Table 1b: Radiology findings of PLCO and NLST patients for the entire data set and stratified by CXR-Lung-Risk groups. For statistical comparison the chi-square test was calculated.

**Table 2 | Patient demographics and clinical risk factors of BLCS patients for the entire data set stratified by CXR-Lung-Risk groups**

| Variables | BLCS | | | | p |
|---|---|---|---|---|---|
| | Entire dataset | Lung-Risk < 65 year | Lung-Risk ≥ 65-<75 years | Lung-Risk > 75 year | |
| *N* | 407 | 89 | 231 | 87 | |
| Age (years) | 65.9 ± 10.7 | 58.3 ± 8.8 | 66.4 ± 10.3 | 72.5 ± 8.6 | <0.001 |
| CXR-Lung-Risk | 70.5 ± 6.7 | 62.9 ± 1.9 | 69.7 ± 2.7 | 80.4 ± 5.4 | <0.001 |
| Sex (m) | 50.1% (204/407) | 33.7% (30/89) | 52.4% (121/231) | 60.9% (53/87) | <0.001 |
| Race | | | | | |
| White | 90.4 (368/407) | 87.6% (78/89) | 91.8% (212/231) | 89.7% (78/87) | 0.22 |
| Black | 4.7% (19/407) | 4.5% (4/89) | 3.5% (8/231) | 8.0% (7/87) | |
| Other | 4.9% (20/407) | 7.9% (7/89) | 4.8% (11/231) | 2.3% (2/87) | |
| Obese (>30 kg/m2) | 25.1% (102/407) | 28.1% (25/89) | 27.3% (63/231) | 16.1% (14/87) | 0.09 |
| Smoking | | | | | |
| Never | 11.5% (47/407) | 19.1% (17/89) | 8.7% (20/231) | 11.5% (10/87) | 0.08 |
| Former | 48.4% (197/407) | 47.2% (42/89) | 47.6% (110/231) | 51.7% (45/87) | |
| Current | 40.0% (163/407) | 33.7% (30/89) | 43.7% (101/231) | 36.8% (32/87) | |
| Tumor Stage | | | | | |
| I | 40.0% (163/407) | 44.9% (40/89) | 39.4% (91/231) | 36.8% (32/87) | 0.45 |
| II | 14.3% (58/407) | 13.5% (12/89) | 12.6% (29/231) | 19.5% (17/87) | |
| III | 45.7% (186/407) | 41.6% (37/89) | 48.1% (111/231) | 43.7% (38/87) | |
| Treatment | | | | | |
| Single treatment regime | 36.9% (150/407) | 36.0% (32/89) | 36.8% (85/231) | 37.9% (33/87) | 0.96 |
| Multimodal regime | 63.1% (257/407) | 64.0% (57/89) | 63.2% (146/231) | 62.1% (54/87) | |
| Lung cancer-related mortality | 42.8% (174/407) | 32.6% (29/89) | 41.6% (96/231) | 56.3% (49/87) | 0.005 |
| Follow-up in years (IQR) | 3.4 (1.5-7.2) | 5.6 (2.8–6.4) | 3.4 (1.4-7.0) | 1.9 (1.1–4.2) | <0.001 |

*BLCS* Boston Lung Cancer Study, *IQR* interquartile range.
As appropriate, the chi-square test, student's t-test, Wilcoxon test or Kruskal Wallis test were calculated. All *p* values are two-sided.

To test whether CXR-Lung-Risk adds incremental value to a baseline multivariable model with the same covariates but without CXR-Lung-Risk, nested Cox proportional hazard models were compared. Adding CXR-Lung-Risk to the baseline model resulted in a modest improvement to estimate lung disease mortality compared to the baseline model alone (c-index: 0.83 [95% CI 0.81–0.85] vs. 0.81 [95% CI 0.79–0.83]).

To account for the confounding effect of smoking, we stratified the PLCO dataset by smoking status ($n = 5505$ ever smoker and $n = 4650$ never smokers). After adjustment for the same risk factors, CXR-Lung-Risk remained independently associated with lung disease mortality in both subpopulations (Supplementary Fig. 3b, c, Supplementary Table 1).

In addition, stratified analyses by sex and chronological age (<65 years old vs. ≥65 years old) are provided in the Supplements (Supplementary Figs. 4 and 5), which revealed similar results for all investigated subgroups.

In a sensitivity analysis, we tested the association between CXR-Lung-Risk and lung cancer-specific mortality in the entire PLCO testing data set and in current or former smokers (quit <15 years ago) with a smoking history of ≥30 pack years to allow for a better comparison to individuals enrolled in NLST (see below). CXR-Lung-Risk showed a graded and independent association with lung cancer-specific mortality after adjusting for demographics and clinical risk factors (Supplementary Fig. 6a, b).

External testing in heavy smokers participating in NLST: As in the PLCO testing data sets, Kaplan-Meier survival curves showed a graded association between CXR-Lung-Risk categories and lung disease mortality in NLST (Fig. 2b). Univariable hazard ratios for lung disease mortality for those with a CXR-Lung-Risk between 65 and 75 years old was 3.03 [2.34–3.93]; $p < 0.001$ and 10.92

[8.07–14.77]; $p < 0.001$ for those >75-year-old. Multivariable hazard ratios adjusted for the same baseline demographics and clinical risk factors as in PLCO were 2.48 [1.88–3.29]; $p < 0.001$ for those with a CXR-Lung-Risk between 65–75 years and 6.48 [4.52–9.31]; $p < 0.001$ for those with a CXR-Lung-Risk > 75-year-old. In addition, CXR-Lung-Risk showed a modest improvement in estimating lung disease mortality when added to the multivariable model of demographics and clinical risk factors alone (c-index: 0.76 [95% CI 0.74–0.78] vs. 0.72 [95% CI 0.70–0.74).

As for PLCO, similar results were seen in NLST participants for sex and age-stratified analyses (Supplementary Figs. 7 and 8) and lung cancer-specific mortality for the entire cohort (Supplementary Fig. 6c).

Testing in patients with early-stage lung cancer from the BLCS: Similar to asymptomatic screening individuals, CXR-Lung-Risk showed a significant graded association with lung cancer-specific mortality in patients with histologically confirmed early-stage (I-III) lung cancer in the BLCS (Fig. 3A). The univariable hazard ratio for a CXR-Lung-Risk between 65–75 years was 1.74 [1.15–2.64]; $p = 0.009$ and 3.30 [2.07–5.25]; $p < 0.001$ for a CXR-Lung-Risk >75 years. After multivariable adjustment for age, sex, race, obesity, smoking status, cancer stage, and treatment, the association for the CXR-Lung-Risk category 65–75 years was attenuated (hazard ratio: 1.28 [0.81–2.02]; $p = 0.30$) but remained robust for those categorized as being >75 years old (hazard ratio: 2.33 [1.36–3.99]; $p = 0.002$). Likewise to the other testing data sets, a small improvement to estimate lung cancer-specific mortality was found for CXR-Lung-Risk when comparing multivariable nested Cox models with and without CXR-Lung-Risk (c-index: 0.76 [95% CI 0.72–0.80] vs. 0.75 [95% CI 0.71–0.79]).

In subanalysis stratified by chronological age (<65 years old vs. ≥65 years old), we found a significant association between CXR-Lung-Risk categories and lung cancer-specific mortality in chronologically

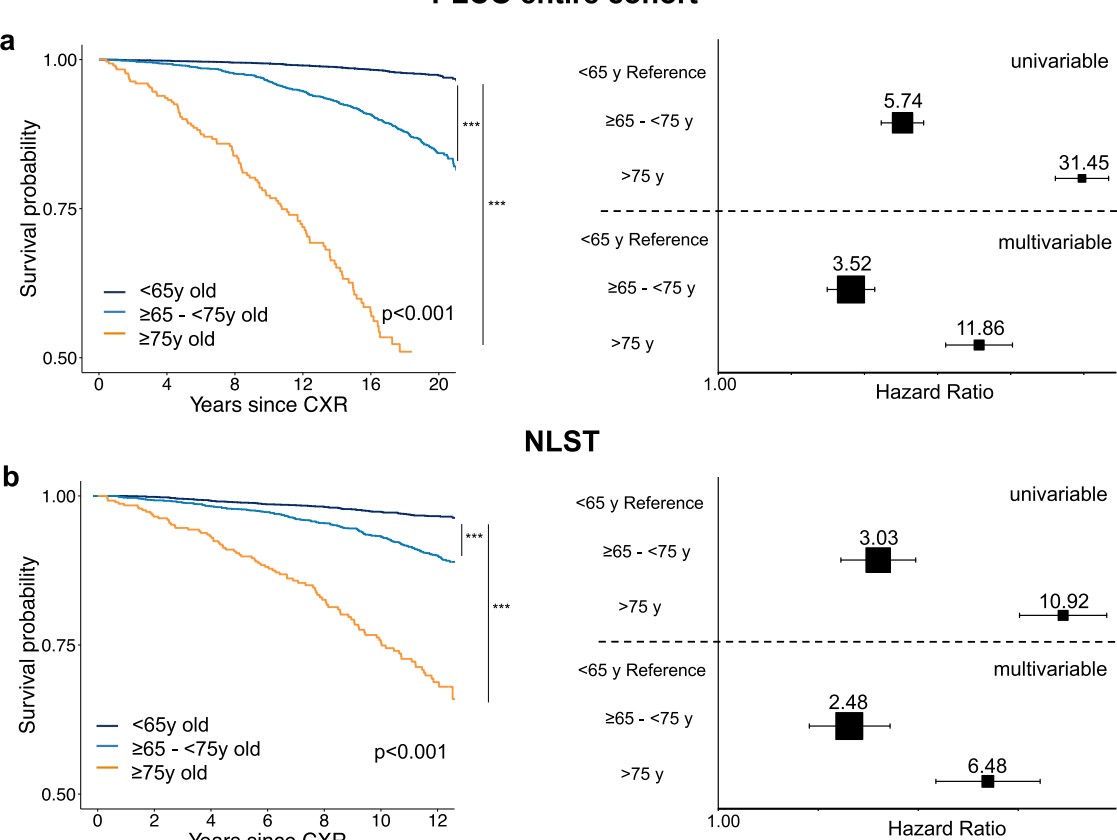

**Fig. 2 | Independent testing of the CXR-Lung-Risk model in the PLCO testing dataset and in NLST to estimate lung disease mortality.** The CXR-Lung-Risk model was independently tested in (**a**) the PLCO testing set ($n = 10155$ independent individuals) and in (**b**) NLST ($n = 5414$ independent individuals). Kaplan-Meier survival analysis shows a graded association between CXR-Lung-Risk groups and lung disease mortality. Pairwise comparison of survival curves was performed using two-sided Log-Rank tests. P-values are adjusted for multiple comparisons using the Bonferroni-Holm method. Forest plots show univariable and multivariable-adjusted hazard ratios (box) with 95% confidence intervals (error bars) for the different CXR-Lung-Risk groups. Multivariable models are adjusted for: chronological age, sex, race, smoking status, pack years, body mass index, prevalent diabetes mellitus, hypertension, history of stroke, myocardial infarction, cancer and 9 chest x-ray findings as described in the methods. Source data are provided as a Source Data file. ***$p$ values $<2*10^{-16}$. CXR chest radiograph, PLCO Prostate, Lung, Colorectal, Ovarian Cancer Screening Trial, NLST National Lung Screening Trial; y years.

older patients, which was not observed in chronologically younger patients (Fig. 3b, c).

To investigate the potential clinical impact of CXR-Lung-Risk in patients with lung cancer we calculated risk reclassification tables based on the CXR-Lung-Risk categories and chronological age (<65 years old vs. ≥65 years old) (Table 3). We found increasing mortality rates by CXR-Lung-Risk categories in both those <65 years of chronologic age and ≥65 years.

In a subset of BLCS patients with available lung function testing ($n = 348$), the proposed CXR-Lung-Risk was compared to a previously described method to estimate a lung age (Lung-Age) via a linear regression using the forced expiratory volume in the first second (FEV1), sex and height[26], which showed a modest correlation (Pearson´s r = 0.45; $p < 0.001$) with CXR-Lung-Risk (Supplementary Fig. 9a). Univariable and multivariable hazard ratios for CXR-Lung-Risk and Lung-Age are provided in Supplementary Fig. 9b, c.

Finally, the relation between CXR-Lung-Risk and FEV1 was investigated, which showed a moderate negative correlation (Pearson´s r = −0.30; $p < 0.001$; Supplementary Fig. 10a). When adding FEV1 to a multivariable model with the same demographic and clinical risk factors as above, the association between CXR-Lung-Risk and lung-cancer-specific mortality remained significant for the >75 years old category (hazard ratio: 1.99 [1.07–3.68]; $p = 0.03$; Supplementary Fig. 10b).

## Discussion

In this study, we propose a deep-learning convolutional neural network that estimates the risk of lung disease mortality from a chest radiograph image as the only input. In three independent testing datasets, CXR-Lung-Risk discriminated individuals at high vs. low risk for lung disease mortality. In addition, CXR-Lung-Risk proved to be independent of and additive to baseline demographics (including age and smoking status), cardiovascular risk factors and traditional radiologic findings after multivariable adjustment. We observed a graded association between CXR-Lung-Risk and individual risk profiles. Higher CXR-Lung-Risk estimates were associated with risk factors like smoking, hypertension, a history of myocardial infarction and stroke as well as traditional radiologic findings. The lowest mean CXR-Lung-Risk (63.0 ± 5.5 years) was found in PLCO, an asymptomatic screening population without known lung cancer 23. NLST25 (all ≥30 pack year smokers) had a higher CXR-Lung-Risk on average (mean 66.1 ± 5.7 years), while the highest CXR-Lung-Risk was found in BLCS patients with histologically confirmed lung cancer (mean 70.5 ± 6.7 years). These findings can be intuitively understood - increasing damage to the chest is associated with higher CXR-Lung-Risk regardless of other risk factors.

These findings could have clinical implications for the treating physician and the patient as decisions on treatment allocation are strongly based on clinical risk factors such as chronological age, FEV1 and comorbidities of an individual to estimate eligibility and

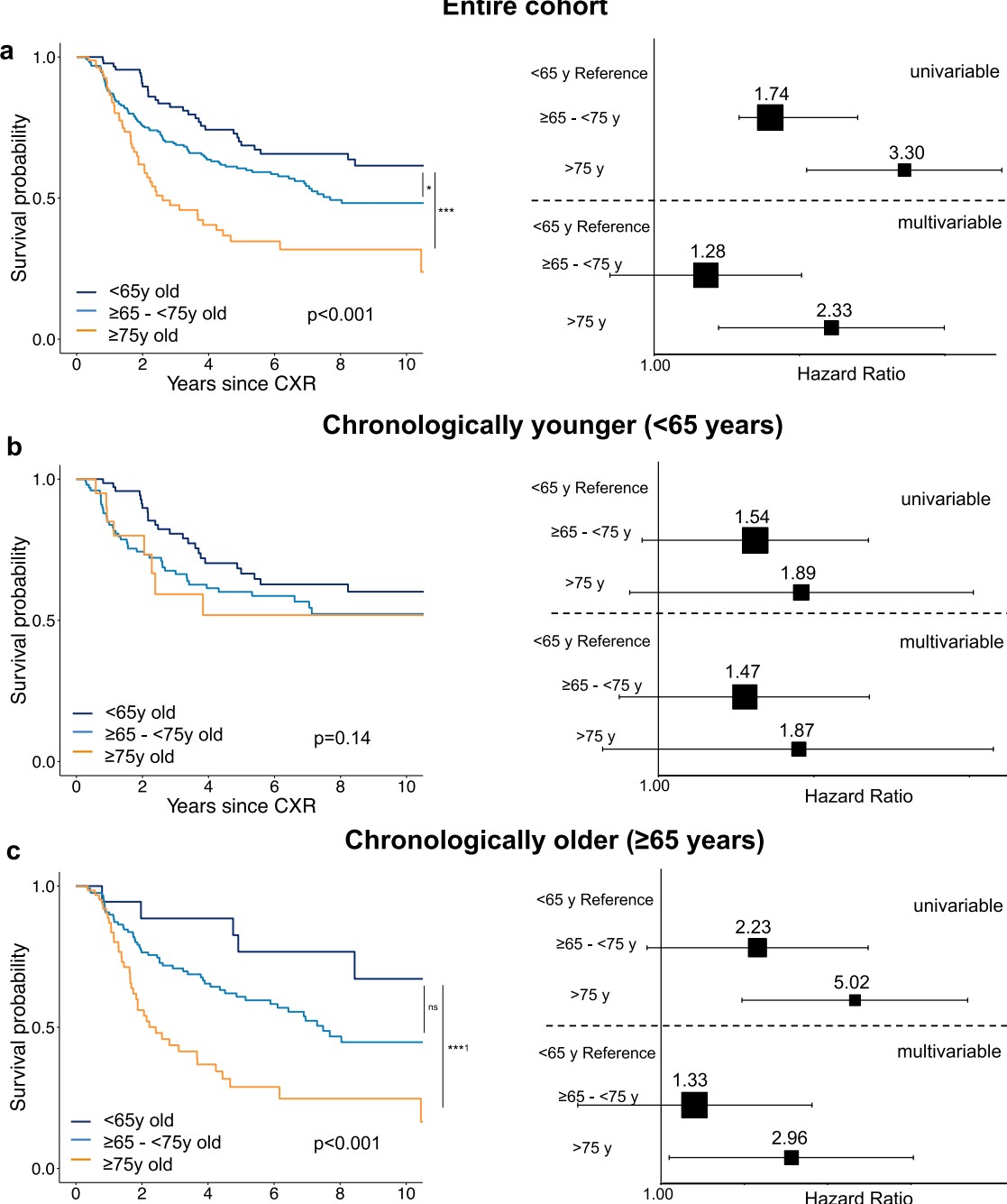

**Fig. 3 | Independent testing of the CXR-Lung-Risk model in the Boston Lung Cancer Study (BLCS) to estimate lung cancer-specific mortality.** In contrast to PLCO and NLST, cause of death was only available for lung cancer but not for other lung diseases. **a** Kaplan-Meier survival analysis shows a graded association between CXR-Lung-Risk groups and lung cancer-specific mortality in the entire cohort ($n = 407$ independent individuals). Subgroup analyses stratified by chronological age (**b**) <65 years old ($n = 194$ independent individuals) vs. (**c**) ≥65 years old ($n = 213$ independent individuals) revealed that this effect seems to be driven by older patients with a chronological age ≥65 years. Pairwise comparison of survival curves was performed using two-sided Log-Rank tests. *P* values are adjusted for multiple comparisons using the Bonferroni-Holm method. Forest plots show univariable and multivariable-adjusted hazard ratios (box) with 95% confidence intervals (error bars) for the different CXR-Lung-Risk groups. Multivariable models are adjusted for chronological age, sex, race, obesity, smoking status, cancer stage, and treatment. Source data are provided as a Source Data file. *$p$ value = 0.01; ***$p$ value = $2.6*10^{-7}$; ns = nonsignificant; ***1 $p$ value = 0.0006; CXR chest radiograph, BLCS Boston Lung Cancer Study, y = years.

tolerability to the chosen regimen[8,27,28]. For example, Walter et al. found that increasing age was negatively associated with the receipt of cancer-directed treatment (e.g. surgery or radiotherapy) in a cohort of more than 13,000 lung cancer patients[29]. Wang et al. reported in a study including more than 20,000 veterans with lung cancer that higher age was a stronger predictor for not receiving guideline-

recommended treatment than the presence of comorbidities[30]. If CXR-Lung risk, a personalized risk estimate, rather than chronological age was used, this could affect treatment decisions for tumor-directed therapy. In this context, CXR-Lung-Risk might be a helpful objective decision-making tool to reduce age-related bias by the treating physician and the risk of withholding a potentially beneficial therapy in an

**Table 3 | Risk reclassification of lung cancer-specific mortality based on risk categories defined by chronological age and CXR-Lung-Risk**

|  |  | CXR-Lung-Risk | | |
| --- | --- | --- | --- | --- |
|  |  | <65 years | ≥65 years – <75 years | ≥75 years |
| Chronological Age | <65 years | 24 (33.8%) | 42 (40.8%) | 9 (45%) |
|  | ≥65 years | 5 (27.8%) | 54 (42.2%) | 40 (59.7%) |

older, but physiologically fit patient[31]. Furthermore, the substitution of chronological age with the biological age captured by CXR-Lung-Risk in existing risk calculators/predictors may provide more accurate decision support. For example, lung cancer prediction models (either for selection of screening candidates or cancer risk prediction of solitary pulmonary nodules) commonly include chronological age[32], and substitution with CXR-Lung-Risk may improve the utility and accuracy of these clinical risk predictions and help for risk reclassification beyond current methods.

Chest radiographs are the most common imaging test and are especially common in persons at risk for lung disease[9]. Although most chest radiographs do not show findings that require clinical interventions, there is increasing evidence that chest radiographs carry additional prognostic information beyond traditional diagnostic findings (e.g. lung consolidations or nodules). For example, in our previous work we demonstrated that deep learning can identify heavy smokers at high risk for incident lung cancer and that a deep learning biological chest x-ray age predicts longevity beyond chronological age and independent of baseline risk factors[13,14]. CXR-Lung-Risk is a new model that allows for identifying individuals at increased risk for mortality of various lung diseases demonstrating that potentially relevant prognostic information captured in a chest radiograph may go unreported. Implementing a tool like CXR-Lung-Risk into the EMR or PACS to automatically extract this currently unused information could help to increase the diagnostic value of this imaging test. For example, only around 5% of eligible Americans are screened for lung cancer[33–35]. Here, the proposed model could help to flag individuals at high risk to prompt risk discussion and encourage entry intro screening programs. Furthermore, it has been reported that approximately 70% of individuals with COPD are underdiagnosed[36] with the risk for increased morbidity and mortality. In this context, CXR-Lung-Risk could be deployed to automatically notify the treating physician to schedule a follow-up visit for the patient to investigate possible causes and discuss potential interventions, such as a full pulmonary function test and the use of bronchodilator. As no human input is necessary, CXR-Lung-Risk could be used with minimal disruption of current clinical workflows and automatically analyze the latest radiograph of a patient at high speed and low additional cost[37]. As such, CXR-Lung-Risk could serve as an early warning system to triage patients into existing screening and chronic pulmonary disease pathways, and to both provide more accurate risk assessments for those programs and increase adherence to guidelines-based therapies.

The following limitations of our study need to be considered. First, the input to the model is a raw chest radiograph. It remains unknown, which alterations and findings in the image are important for the final prediction. This is a common drawback of deep learning models that may limit the acceptance by physicians and patients to use this information for clinical decision-making. However, association analysis shows correlation with clinical risk factors (e.g. smoking, age, prevalent hypertension) and traditional radiologic findings (e.g. nodules, fibrosis, emphysema) suggesting that the model identifies anatomical changes known to be correlated with increased risk. Second, the majority of participants in all datasets (development and testing) were Non-Hispanic White. Detailed analysis regarding generalizability of the model to other races and ethnic groups was not

possible in our datasets and needs to be investigated in future studies. Third, testing CXR-Lung Risk in BLCS as a potential clinical use case using existing chest radiographs obtained through routine care only comprised a relatively small hospital cohort of lung cancer patients. Whether there is a similar prognostic value for early detection/prognosis of other lung diseases such as COPD or asthma or even broader adoption remains to be seen. Fourth, the age range in PLCO was 55–74 years old, which will likely limit the value of the model in substantially younger individuals. Moreover, although CXR-Lung Risk accurately stratified risk in lung cancer patients, it remains unknown whether this improves clinical decision-making or treatment planning. This needs to be tested in future prospective trials. In addition, many patients at increased risk for lung cancer or prevalent disease get other imaging tests, including serial computed tomography. Whether specifically tailored models to estimate prognosis using this imaging data needs to be investigated in additional studies. Further, in PLCO and NLST there is a discrepancy between the relatively small increase in the c indices in nested model comparison and the large hazard ratios, especially in the high-risk groups (CXR-Lung-Risk >75 years), which is likely explained by the significantly different number of individuals in the different risk groups. Finally, PLCO chest radiographs were collected from 1993-2001 and available as scanned films. Whether this has an impact on model accuracy in more modern datasets was not systematically analyzed in the current study. However, independent testing in BLCS, where the most recent radiographs were acquired in 2016, showed robust performance.

In conclusion, a deep learning model can estimate risk of lung disease mortality from a chest radiograph beyond demographics, including smoking status, cardiovascular risk factors and traditional radiologic findings and may help to identify high-risk individuals in screening and cancer populations.

## Methods

All analyses performed in this study comply with relevant ethical regulations. Secondary use of the PLCO, NLST and BLCS cohorts has been approved by the Mass General Brigham, Boston, Massachusetts institutional review board. All participants provided informed consent at enrollment into the original study.

The CXR-Lung-Risk model was developed in a large multicenter prospective cancer screening trial and independently tested in one internal and two external, held-out datasets not seen during any part of the development process. Results are reported for the three testing data sets only. An overview of the study design is provided in Fig. 1.

### Model development

The CXR-Lung-Risk model was developed using data from the Prostate, Lung, Colorectal, Ovarian (PLCO) Cancer Screening Trial[23,24], as it was the largest available dataset in the current study. PLCO was a multicenter randomized controlled trial of chest radiography for cancer screening in asymptomatic individuals aged 55–74 years enrolled at 10 US sites from 1993 through 2001. Individuals in the intervention arm received a chest radiograph at enrollment and up to 3 annual follow-up radiographs. For model development, a random sample of 80% ($n = 40,643$) of individuals enrolled in the intervention arm was used, including chest radiographs from all timepoints ($n = 147,497$). 20% of the training data was reserved for hyperpameter tuning. For model development, each radiograph exam was used as an independent sample; for testing, only baseline radiographs defined as the initial radiograph obtained at the enrollment (T0) exam were used. The only input to the proposed CXR-Lung-Risk model is a chest radiograph image; the output is an estimated risk of 18-year lung disease mortality (defined below) expressed in years (e.g., CXR-Lung-Risk of 75 years means an equal risk of lung disease-related death as the average 75-year-old individual). Usually, risk probabilities are expressed in percentages, which are difficult to grasp. Therefore, we decided

to express CXR-Lung-Risk in years rather than a probability between 0-100%. In contrast to our previous work[15], which was developed as a single prediction model, CXR-Lung-Risk was built as an ensemble model to reduce variance in the output[38,39]. The ensemble consisted of 20 CNNs. The model architecture for each CNN was chosen randomly from a set of architectures popular in medical image analysis (inceptionv4, resnet34, tiny[40–42]). Hyperparameters for each model were randomly selected during training (Supplementary Table 2), as random hyperparameters have been shown to improve performance in ensemble learning by reducing correlation between models[43]. The output of these 20 models was combined into a single prediction using a LASSO regression model trained on the hyperparameter tuning dataset. LASSO regression coefficients are given in Supplementary Table 2. We found that 13 out of 20 models had a nonzero LASSO regression coefficient and were included in the final ensemble. A comparison of these 13 single models vs. the ensemble model is given in Supplementary Fig. 11.

Instead of a binary target variable of lung-related mortality, we defined age-adjusted labels reflecting the risk of lung disease mortality based on prevalent risk factors for those that did not die of a lung-related disease during follow-up. We posit that these labels are more informative than assigning a "0" for all individuals that did not die, regardless of their underlying risk profile. We define these age-adjusted labels according to the following equation:

$$LR = CA + (E - D)$$

where $LR$ is the Lung-Risk label, $CA$ is the current chronologic age, $E$ is expected age-at-death based on US social security life tables[44]. $D$ corresponds to age-at-death based on A) actual, observed age at death for those that died of a lung disease or lung cancer or B) an individual's predicted age at death due to lung disease/lung cancer based on a survival regression model trained using data from the control (no imaging) arm of the PLCO trial ($n = 77,444$) for those who did not die (Supplementary Table 4). This regression model used prevalent risk factors as input to estimate the age an individual would die of lung-related disease. This model accurately estimated age at death due to lung disease with a concordance index of 0.82 (95% CI [0.817–0.825]) for all lung disease mortality and 0.84 (95% CI [0.835–0.845]) for lung cancer death. This approach was similar to our previous work in which we trained a model to estimate a general biological chest x-ray age[14]. Cause of death was adjudicated by the trial based on death certificates and the National Death Index. These Lung-Risk labels were only used for training. The reported results in the testing datasets are all based on the actual observed deaths during follow-up.

For PLCO and NLST participants, only the baseline radiograph was used, which was acquired upright in posterior-anterior projection. PLCO radiographs were provided as scanned films in .tif file format with protected health information redacted using black pixels. PLCO radiographs were converted to Portable Network Graphics (.png) format using ImageMagick v6.8.9-9. For NLST, all chest radiographs were available in Digital Imaging and Communications in Medicine (DICOM) format. Radiographs for BLCS patients were acquired through clinical care and available in DICOM format from the hospital's Picture Archiving and Communications System. We converted NLST and BLCS DICOM files to .tif using DCMTK v3.6.1 and then to .png using ImageMagick to maintain consistency with the aforementioned PLCO radiographs. To increase the sample size in BLCS, both posterior-anterior or anterior-posterior radiographs were included if they were taken up to 3 months prior to histologically confirmed lung cancer diagnosis. For training, images were rescaled to 224 pixels on the short-axis and randomly cropped to 224 ×224 before input into the model. This random cropping was done each time the image was fed to the model as a form of data augmentation. Additional data augmentations used for training included mixup data augmentation, up to 20

degrees of random rotation, up to 20% zoom in/out, and up to 40% brightness/contrast adjustments.

The model was trained using a mean-squared error loss function with the ADAM optimizer. The number of epochs for training was selected uniformly at random between 40 and 70. This was independently chosen for each of the 20 models. Training was done using an Ubuntu Linux workstation with an AMD 3960×24-core CPU with 128 GB of system RAM, and a single NVIDIA RTX A6000 GPU with 48 GB of GPU RAM. The model was developed using fastai v2.5.3, PyTorch v1.10, and CUDA v11.2. The full source code and programming environment are freely available at https://aim.hms.harvard.edu/cxr-lungrisk.

The final model was an ensemble of 20 models combined using a LASSO regression. LASSO regression coefficients were fit using the glmnet package in R, and hyperparameters for the LASSO were selected based on the minimum mean-squared error in an internal 10-fold cross-validation. LASSO regression coefficients for each of the 20 models are given in Supplementary Table 2 and the performance of each of the 20 models in Supplementary Fig. 11.

Model testing: The CXR-Lung-Risk model was tested in three independent test datasets not used during any part of model development. All results are reported for the testing datasets only based on the actual observed lung diseases mortality. The first testing dataset comprised a random sample of 10,155 asymptomatic individuals from PLCO aged 55–74 (20% of individuals; median follow-up=17.0 [IQR 14.8–19.0] years) not used for model development[23]. Annual questionnaires, communication with next of kin and the National Death Index were used to determine mortality. Cause of death was defined based on International Classification of Disease-9 (ICD) codes (Supplementary Table 3) (clinical trial registration number: NCT00047385). For model testing, only baseline radiographs were used.

The second testing dataset included 5,414 participants from the chest radiograph arm of the National Lung Screening Trial (NLST)[25]. NLST was a randomized controlled trial that enrolled heavy smokers (≥30 pack years) aged 55–74 years for lung cancer screening via chest CT vs. chest radiograph at 21 US sites from 2002 through 2004. Each participant had a baseline scan and up to 2 annual follow-up scans if no lung cancer was detected. Median follow-up time was 11.9 [IQR 7.3–12.3] years. Mortality was assessed via annual questionnaires, communication with next of kin and the National Death Index. Cause of death was determined using ICD-9 codes (Supplementary Table 3) (clinical trial registration number: NCT01696968). For this study, only the baseline chest radiograph was used.

The third testing dataset included 407 patients from the Boston Lung Cancer Study (BLCS), which is an ongoing multicenter observational epidemiologic cohort registry of patients with histologically confirmed lung cancer. For this study, only patients with early-stage (I-III) and a diagnosis between 2004-2016 were included. Median follow-up was 3.4 [IQR 1.5-7.2] years. Death was verified by dedicated study personnel via manual chart review. In contrast to PLCO and NLST, cause of death was only collected for lung cancer but not for other lung diseases. A consort diagram for all three study cohorts is provided in Supplementary Fig. 12.

Clinical covariates, radiographs and traditional radiographic findings: Baseline demographics and prevalent risk factors such as diabetes, hypertension or smoking status are self-reported by trial participants in PLCO[23] and NLST[25]. For BLCS patients, all clinical covariates were extracted from the electronic medical record by dedicated study staff. Unlike PLCO and NLST, pack-years were not available in BLCS for all individuals and not included in the analysis. Traditional radiographic findings including lung nodules, atelectasis, pleura and lung fibrosis, COPD/emphysema, opacities, cardiac abnormalities, lymphadenopathy and bone/chest wall lesions were only available for PLCO and NLST and reported by centrally qualified radiologists for all participants.

Outcomes: The primary endpoint of this study was a composite of lung disease mortality, including lung cancer, interstitial pulmonary

disease, emphysema, and COPD; the secondary endpoint was lung cancer-specific mortality. All cause-specific deaths were based on ICD-9 codes provided in the PLCO[23] and NLST[25] trials (Supplementary Table 3). For BLCS, endpoints were verified by manual chart review and available for lung cancer-specific mortality only.

Statistical analysis: Continuous variables are presented as mean ±standard deviation (SD) or median and interquartile range (IQR). Categorical variables are reported as frequencies and percentages. Baseline demographics were compared using the student's t-test or Kruskal-Wallis test for continuous variables, as appropriate. For categorical variables, the Chi-square test was conducted.

To investigate time to lung disease mortality and lung cancer-specific mortality, Kaplan-Meier survival estimates and log-rank tests were calculated. The association between CXR-Lung-Risk and time to lung disease mortality as well as lung cancer-specific mortality was assessed via univariable and multivariable Cox proportional hazards regression analysis. Multivariable models in PLCO and NLST were adjusted for the following covariates: age, sex, race, smoking status, pack years, body mass index, prevalent diabetes mellitus, hypertension, history of stroke, myocardial infarction, and cancer. For BLCS patients, the following covariate were available: age, sex, race, obesity, smoking status, cancer stage (I-III), and treatment (surgery only vs. adjuvant treatment). Additionally, in BLCS patients with available lung function testing ($n$ = 348; FEV (l) = forced expiratory volume in liters in 1 second), lung age as proposed by Morris et al.[26] was calculated (lung $age_{women}$ = 3.56*height − 40 (FEV1) − 77.28; lung $age_{men}$ = 2.87*height − 31.25 (FEV1) − 39.375) and compared to CXR-Lung-Risk proposed in this study. For all cohorts, sex-stratified analyses were performed. All $p$ values are two-sided and considered statistically significant if below 0.05. All statistical analyses were performed in R (version 3.6.1).

### Reporting summary
Further information on research design is available in the Nature Portfolio Reporting Summary linked to this article.

## Data availability
The original PLCO and NLST data cannot be distributed with this publication due to our data use agreements but can be downloaded upon request from the National Cancer Institute (PLCO: https://biometry.nci.nih.gov/cdas/plco/; NLST: https://biometry.nci.nih.gov/cdas/nlst/). The BLCS data are protected under the BLCS study protocol. Access to limited de-identified data can be requested through the BLCS Trial Center for academic non-commercial research purposes only and are subject to review of a project proposal that will be evaluated by a BLCS data access committee. Requests can be made through BLCS webpage (https://www.hsph.harvard.edu/blcs/) or directly by contacting Prof. David Christiani (dchris@hsph.harvard.edu). Requests will be reviewed within two weeks. Source data are provided with this paper including the CXR-Lung-Risk estimates generated in this study, which have been deposited on the AIM webpage (https://aim.hms.harvard.edu/cxr-lungrisk) and are available for download to replicate the statistical analysis. Source data are provided with this paper.

## Code availability
All the code of the deep learning system including the trained model and the code of the statistical analyses is publicly available on the AIM webpage: https://aim.hms.harvard.edu/cxr-lungrisk. Furthermore, we embedded the deep learning model in an end-to-end pipeline including image preprocessing and model inference that is freely available in a Google Colab cloud-based notebook. This cloud-based instance facilitates future validation studies and allows users with minimal coding proficiency to process a large amount of CXR data without having to install anything on their local node. In the notebook, we describe all the steps of the processing, discuss the different models composing the ensemble and their details, and provide examples. This notebook along with R code to reproduce the statistical analyses can also be found on the AIM webpage: https://aim.hms.harvard.edu/cxr-lungrisk.

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

## Acknowledgements

The authors thank the study participants, the investigators, and the NCI for data collected in the PLCO and NLST trials. Original data collection for the ACRIN 6654 trial (NLST) was supported by NCI Cancer Imaging Program grants. The statements contained herein are solely those of the authors and do not represent or imply concurrence or endorsement by the above organizations. The authors further acknowledge financial support from NIH (NIH (NCI) 5U01CA209414, DC; NIH-USA U24CA194354, HA; NIH-USA U01CA190234, HA; NIH-USA U01CA209414, HA; and NIH-USA R35CA22052, HA), the European Union - European Research Council (866504; HA) and the National Academy of Medicine Healthy Longevity Grand Challenge (2000011734; VR, JW, ML).

## Author contributions

Contributions: Study design: J.W., V.R., M.T.L., H.J.W.L.A.; code design, implementation and execution: V.R., D.B.; acquisition, analysis or interpretation of data: J.W., V.R., D.C., R.M., M.T.L., H.J.W.L.A.; writing of the manuscript: J.W., V.R., M.T.L., H.J.W.L.A.; critical revision of the manuscript for important intellectual content: all authors; statistical analysis: J.W., V.R.; study supervision: M.T.L., H.J.W.L.A.

## Competing interests

The authors declare no competing interests.
