## [Peer Review File · Nature Communications]

REVIEWER COMMENTS

Reviewer #1 (Remarks to the Author):

In this study, the authors developed a deep learning model to predict the risk (CXR lung-risk) of lung disease mortality from regular lung radiographs. In three independent test datasets, they found the lung risk to significantly associate with lung disease mortality.

This is an interesting study. The authors found a novel problem for deep learning applications. The evidence seems to suggest that the deep learning models can identify subjects with elevated risk to develop lung disease in the future. However, the reviewer found the manuscript to lack sufficient details in methodology.

Major comments:

- The deep learning model description is vague:
 - o The definition of lung-risk label seems to make sense but is presented as is. The authors are encouraged to provide some rationale.
 - o What's the definition of your loss function? This is completely missing.
 - o There is also a lack of information regarding how the deep learning models are trained. What optimizer is used? How is the development set split? How many epochs are trained?
 - o In the PLCO dataset, each patient contains three exams. Do you use each exam as an independent sample for training? Please clarify.
 - o Briefly describe the regression used to predict ages. Readers shall not be referred to read the authors' previous study. How accurate is the prediction?
 - o The 20 deep learning models were trained with rather random hyperparameters, why is that?
 - o Why do you have to use so many models for ensemble learning? How do each individual model perform? This is unacceptable. Extreme ensemble learning is only used in ML contest to boost performance but rarely used in clinical settings.
 - o How are the 20 models combined? The authors mentioned LASSO but there are insufficient details.
 - o The authors are encouraged to describe their deep learning models clearly in the main text. Such important information shall not be tucked away in the supplement.
 - o The preprocessing of the images is completely missing. How did you convert them from DICOM to PNG? What are the dimensions of the images? Did you use any cropping and/or padding? What data augmentations did you use?

- On line 132, the authors claim, “CXR Lung-Risk remained independently associated with lung disease mortality in both subpopulations”. But in supplemental Fig. 3B, the lung-risk hazard ratio of the [65, 75] category is not significant in the multivariate model. This sentence is misleading.

Some minor comments:

- On line 302, what do you mean by “baseline radiographs”?
- The concept of CXR Lung-Risk shall be introduced in the beginning of the article or else it’s rather confusing.

Reviewer #2 (Remarks to the Author):

This is an interesting manuscript using deep learning applied to routine chest radiographs and showing an ability to predict lung death.

I am not a methodology expert but assume the convolutional neural network analyses are done appropriately. I wonder how different the model had looked if one of the other cohorts had served as generator of the model, though.

I have a number of critical comments.

First, I am not sure how "routine" these radiographs are as they are all part of 3 large trials/studies so I assume there is some underlying standardisation.

Secondly, the population in all three cohorts consist of heavy smokers at high risk of lung cancer or subjects who have already been diagnosed/treated for lung cancer. This is a fairly narrowly defined population and I would assume that in many places those at high risk are now offered CT screening and those with recent lung cancer are likely followed up with frequent and/or more advanced imaging?

I am not convinced that the effects can not simply be explained by residual confounding from smoking and other risk factors. As I read methods, the authors adjust for smoking but not pack-years. Pack-years is associated with outcome and changes such as emphysema are also associated with pack-years and like a factor of importance within the model.

Also, did the authors have access to lung function testing. It would be interesting to see if these variables influenced the model or perhaps weakened the predictive value.

Reviewer #3 (Remarks to the Author):

The paper describes the development and evaluation of a deep learning algorithm to estimate the risk of lung disease mortality using chest radiographs. Overall, the paper is clearly written and the approach seems appropriate. The main questions are whether the improvement in model performance is sufficient to merit broader adoption, which is compounded by question about the ease of clinical application of the proposed tool.

1. Population of interest. The model is developed using the PLCO cancer screening trial data on asymptomatic individuals age 55-74 enrolled at 10 US sites between 1993-2001. The model is validated in a hold-out sample from this population and two other, higher-risk data sets, which is appropriate. One question: was the hold-out sample from PLCO selected chronologically (the newest data) or randomly? Chronological selection is preferred as it better matches clinical reality (we develop model now but apply it to new patients).
2. Outcome of interest. Outcome is defined as lung disease mortality. Has this outcome been captured consistently across the evaluation samples?
3. Time horizon. The 3 studies in question have differing follow-ups with medians ranging between 17, 12 and 3 years. It is not clear from the descriptions in the paper what was the time horizon for predictions. The authors express the risk score in years, having made a conversion from % risk to age of a person with a given level of risk. But level of risk needs a specified time horizon.
4. Predictors. The primary predictor is the CXR Lung-Risk built based on convolutional neural networks (CNN). It was added to the model containing standard risk factors. Valid approach.
5. Mathematical model. CNN for CXR Lung-Risk and Cox PH model for adding CXR Lung-Risk to other risk factors. This is a solid approach.
6. Model performance. While CXR Lung-Risk expressed as age appears to be very strong in terms of hazard ratios, we need to realize that this form of presentation combines the effect of the images with that of age. As the authors explain, a risk of 75 means the risk of a person who is 75 years old. This means the average of all 75-year-olds and those with risk at the level of average 75-year-old. When compared to the risk of 65 which is defined similarly, this combines age and image information. This explains why the very large hazard ratios translate into small increments in the C-indices and leave us with the question to what extent it is worth the effort.
7. Clinical Decision Support. The authors leave us with the hazard ratios and C-indices but do not propose how their model could be used in practice. Should we look at risk reclassification?
8. How are these patients treated today? What can be done for them differently if we had a risk score, one with imaging data?

Reviewer #4 (Remarks to the Author):

In this paper, the authors developed a deep learning model, named CXR Lung-Risk, that predicts the risk of lung disease mortality from a routine CXR image. The deep learning model, which is actually an ensemble of several AI models, has been trained on 80% of the data from the PLCO trial, and is validated on the remaining 20% of the PLCO trial (n=10155), a dataset from the NLST dataset (heavy smokers) (n=5414), and a clinical multi-center dataset from the Boston Lung Cancer Study (n=407). The authors found that the model showed a graded association with lung disease mortality, and this association remained present after adjusting for known risk factors, including chronological age, smoking, and radiologic findings.

The final claim of this manuscript is that the results demonstrate that deep learning can identify individuals at high risk of lung disease mortality from easily obtainable, low-cost, and routine CXR images.

The manuscript is well written, clear, and newsworthy. The paper is related to a previous publication in which the lung age was predicted from a routine chest radiography image using machine learning. The model is externally validated on two datasets (NLST and BLCS). The PLCO and NLST are datasets which are both quite old and therefore, the additional validation on the data from the BLCS study is an important addition. The manuscript contains no evidence about the clinical impact of the CXR Lung-Risk AI model, but this is the topic of future studies.

I have the following comments and points of criticism:

- Please add in what years the CXRs from the BCLS study were obtained in the Methods section.
- There are too little details regarding the training and architecture of the models. In my opinion, more details should be added. What architectures were used specifically? What loss function was used? On what hardware was this trained? etc
- The manuscript is inconsistent in the terms multivariable vs multivariate. The methods section mentions multivariable while the rest of the manuscript uses multivariate. What is correct? Please check carefully and adjust.
- This claim in the Discussion is too strong and without evidence: "If CXR-Lung risk, a personalized risk estimate, rather than chronological age was used for determining the type of treatment, these patients would have significantly more likely received a tumor-directed therapy. "

Please adjust.

- What about the quality of the chest radiographs? The model is trained with radiographs acquired between 1993 and 2001. Do the authors expect problems with data drift when this model is applied in more modern datasets? I think this needs discussion.

- Please add subgroup analysis for different age groups on PLCO and NLST as well in the manuscript, as was done for the BLCS study.

General Response to the reviewers

We thank the reviewers for their detailed feedback and have responded in blue. We would like to address common comments here:

- **Inclusion of sufficient technical details:** We agree with the referees that relevant technical details about our methodology have to be included in the manuscript. We have addressed this important point in the revision and added detailed information including the model development procedure with information about the architecture, the loss function, the optimizer, the number of epochs for training, and the hardware/software environment. In addition, important details such as the definition of the training labels, data training splits, and hyperparameters search have been included as well, which can be found in the revised Methods and Supplements of the manuscript.
- **Sharing of Code:** In support of open science and transparency, we have now implemented and will publicly share an end-to-end reproducible pipeline including image pre-processing code, model architecture and weights, model inference code, and statistical analysis code. This will ensure the full source code and programming environment will be made publicly available upon publication. For review, a zip file with all the code is also provided with this submission. Additionally, we will host a dedicated website for this platform on our homepage and support questions from the community: <https://aim.hms.harvard.edu/cxr-lungrisk>. Please also see the revised Data Sharing and Code Sharing statements in the revised manuscript.
- **Additional Experiments:** Based on the reviewer's suggestions we also performed new experiments relevant to our study. These include investigating the relationship between CXR Lung-Risk and FEV1, a functional lung test. We have added correlation analysis and Cox regression analysis including FEV1 in the revised manuscript, showing that CXR Lung-Risk predictions remain significant after adjustment for FEV1. Furthermore, we included new experiments demonstrating the superiority of our ensemble classifier approach over single models. Finally, we also included new experiments in stratified analyses by sex and chronological age, which revealed similar results for all investigated subgroups. More details can be found in the updated Results and Supplements sections.

The revised manuscript was updated to incorporate these important new insights and data. For more details, please find our point-by-point response below.

Specific Response to the Reviewers' Comments

Reviewer #1 (Remarks to the Author):

In this study, the authors developed a deep learning model to predict the risk (CXR lung-risk) of lung disease mortality from regular lung radiographs. In three independent test datasets, they found the lung risk to significantly associate with lung disease mortality.

This is an interesting study. The authors found a novel problem for deep learning applications. The evidence seems to suggest that the deep learning models can identify subjects with elevated risk to develop lung disease in the future.

We would like to thank the reviewer for these positive remarks.

However, the reviewer found the manuscript to lack sufficient details in methodology.

We agree and apologize for omitting important details about our methodology. We have addressed this important point in the revision and added many details about the methodology in the revised manuscript. Furthermore, we have now implemented and will publicly share an end-to-end reproducible pipeline including image pre-processing code, model architecture and weights, model inference code, and statistical analysis code, upon publication. This will ensure the full source code and programming environment will be made publicly available without restrictions. For review, a zip file with all the code is also provided with this submission. Additionally, we will host a dedicated website for this platform on our homepage and support questions from the community: <https://aim.hms.harvard.edu/cxr-lungrisk>. Please also see the revised Data Sharing and Code Sharing statements in the revised manuscript.

Major comments:

Comment 1.1: The deep learning model description is vague:

We apologize for omitting important details about our model development procedure and have addressed this in the revision. For example, we have added information about the loss function, the optimizer, the number of epochs for training, and the hardware/software environment.

Methods page 13, paragraph 3:

"The model was trained using a mean-squared error loss function with the ADAM optimizer. The number of epochs for training was selected uniformly at random between 40 and 70. This was independently chosen for each of the 20 models. Training was done using an Ubuntu Linux workstation with an AMD 3960x 24-core CPU with 128 GB of system RAM, and a single NVIDIA RTX A6000 GPU with 48 GB of GPU RAM. The model was developed using fastai v2.5.3, PyTorch v1.10, and CUDA v11.2. The full source code and programming environment are freely available at <https://aim.hms.harvard.edu/cxr-lungrisk>"

Comment 1.2: The definition of lung-risk label seems to make sense but is presented as is. The authors are encouraged to provide some rationale.

The rationale for the lung-risk label is to provide the model with more informative labels than just using a binary (0 or 1) outcome. Those that died of a lung-related disease during follow-up were assigned higher-risk labels. For those that did not die of a lung-related disease, we assigned higher risk labels to those with more prevalent risk factors (e.g., age, smoking, history of disease).

As stated in the Methods, Page 11, Paragraph 2 we chose to express the labels in terms of an “age” in years instead of a risk probability to make the output more intuitive. These lung-risk labels were used during model development only. For testing the model, we used an outcome of observed lung-related mortality. The only input to the proposed CXR Lung-Risk model is a routine chest radiograph image; the output is an estimated risk of 18-year lung disease mortality expressed in years (e.g., CXR Lung-Risk of 75 years means an equal risk of lung disease-related death as the average 75-year-old individual). Usually, risk probabilities are expressed in percentages, which are difficult to grasp. Therefore, we decided to express CXR Lung-Risk in years rather than a probability between 0-100%.

Methods, page 13, paragraph 2:

“Instead of a binary target variable of lung-related mortality, we defined age-adjusted labels reflecting the risk of lung disease mortality based on prevalent risk factors for those that did not die of a lung-related disease during follow-up. We posit that these labels are more informative than assigning a “0” for all individuals that did not die, regardless of their underlying risk profile.”

Comment: 1.3: What’s the definition of your loss function? This is completely missing.

We agree with the reviewer and have added this important point to the revised manuscript (see the response to Comment 1.1).

Comment: 1.4: There is also a lack of information regarding how the deep learning models are trained. What optimizer is used? How is the development set split? How many epochs are trained?

We apologize for leaving out these important details. We used the ADAM optimizer and trained each model for 40-70 epochs (see Comment 1.1). We used 20% of development set participants for hyperparameter tuning and the remaining 80% of participants for training. We have clarified this in the revised Methods.

Methods, page 12, paragraph 2:

“For model development, a random sample of 80% (n=40,643) of individuals enrolled in the intervention arm was used including chest radiographs from all time points (n=147,497). 20% of the training data was reserved for hyperparameter tuning. For model development, each radiograph exam was used as an independent sample; for testing, only baseline radiographs defined as the initial radiograph obtained at the enrollment (T0) exam were used.”

Comment 1.5: In the PLCO dataset, each patient contains three exams. Do you use each exam as an independent sample for training? Please clarify.

Yes, we used each exam as an independent sample for training; however, only baseline radiographs (1 per participant obtained at enrollment) were used for testing. Please see our previous response (Response 1.4) on how we updated the revised manuscript accordingly.

Comment 1.6: Briefly describe the regression used to predict ages. Readers shall not be referred to read the authors’ previous study. How accurate is the prediction?

The regression model used prevalent risk factors (age, sex, smoking, etc) to estimate the age an individual would die of a lung-related disease. The model was trained on participants from the “control” arm of the PLCO trial (N=77,444). The goal of the model is to predict the age that an individual would die of a lung-related disease or lung cancer. We assessed the accuracy of this

regression model by calculating the c-statistic when applying the model to the “imaging” arm of the PLCO study. We added these performance metrics to the Methods:

Methods, page 13, paragraph 2:

“We define these age-adjusted labels according to the following equation:

$$LR = CA + (E - D)$$

where LR is the Lung-Risk label, CA is the current chronologic age, E is expected age-at-death based on US social security life tables⁴⁴. D corresponds to age-at-death based on A) actual, observed age at death for those that died of a lung disease or lung cancer or B) an individual's predicted age at death due to lung disease/lung cancer based on a survival model trained using data from the control (no imaging) arm of the PLCO trial (n=77,444) for those who did not die (Supplementary Table 4). This regression model used prevalent risk factors as input to estimate the age an individual would die of lung-related disease. This model accurately estimated age at death due to lung disease with a concordance index of 0.82 (95% CI [0.817 – 0.825]) for all lung disease mortality and 0.84 (95% CI [0.835 – 0.845]) for lung cancer death. This approach was similar to our previous work in which we trained a model to estimate a general biological chest x-ray age¹⁴. Cause of death was adjudicated by the trial based on death certificates and the National Death Index. These Lung-Risk labels were only used for training. The reported results in the testing datasets are all based on the actual observed deaths during follow-up.

We have also added Supplementary Table 4 with the regression coefficients for this model:

Supplementary Table 4:

Sex-specific survival models developed in the control (no radiograph) arm of PLCO (N = 77,444).

Risk factor	Coefficient (Male)	p-value (Male)	Coefficient (Female)	p-Value (Female)
Diabetes	-1.54	<0.001	-3.51	<0.001
Age (per 1 yr)	-0.69	<0.001	-0.68	<0.001
Obese	-0.10	0.75	-0.15	0.72
Underweight	-6.27	<0.001	-7.19	<0.001
Hypertension	-0.65	0.02	-0.97	0.006
Past MI	-2.14	<0.001	-2.91	<0.001
Past Stroke	-1.47	0.03	-3.35	<0.001
History of Cancer	-1.38	0.004	-1.42	0.002

Pack-Years	-0.08	<0.001	-0.12	<0.001
Current smoking	-10.87	<0.001	-11.20	<0.001
Former smoking	-4.07	<0.001	-3.80	<0.001
Intercept	83.58		86.53	

Supplementary Table 4: Models were used to estimate age-at-lung-death for those that did not die of a lung-related disease during follow-up. Models were not used for testing dataset results, where only actual, observed lung-related mortality is reported. For example, expected age of dying of a lung-related disease for a 66 year-old male with hypertension and past-MI (myocardial infarction) would be given by:

$$\text{Current Age} + \text{Intercept for men} - \text{age} * 0.64 - \text{coefficient}(\text{hypertension}) - \text{coefficient}(\text{past MI}) = 66 + 65.48 - (66 * 0.64) - 1.71 - 2.78 = 66 + 18.75 = 84.75$$

Comment 1.7: The 20 deep learning models were trained with rather random hyperparameters, why is that?

In ensemble learning, the aim is to develop individual models that are “diverse” in that their predictions are uncorrelated to one another, so that there is a performance benefit when the models are combined. We used diverse, random hyperparameters to encourage uncorrelated predictions from the 20 base models. We have added this explanation to the Methods.

Methods, page 12, paragraph 2:

“...,CXR Lung-Risk was built as an ensemble model to reduce variance in the output.^{38,39} The ensemble consisted of 20 CNNs. The model architecture for each CNN was chosen randomly from a set of architectures popular in medical image analysis (inceptionv4, resnet34, tiny⁴⁰⁻⁴²). Hyperparameters for each model were randomly selected during training (Supplementary Table 2), as random hyperparameters have been shown to improve performance in ensemble learning by reducing correlation between models.⁴³

43. Wenzel, F., Snoek, J., Tran, D., & Jenatton, R. Hyperparameter Ensembles for Robustness and Uncertainty Quantification. Adv. Neural Inf. Process. Syst. 33, (2020).

Comment 1.8: Why do you have to use so many models for ensemble learning? How do each individual model perform? This is unacceptable. Extreme ensemble learning is only used in ML contest to boost performance but rarely used in clinical settings.

We agree with the reviewer that more information about the model architecture should be included, and we have addressed this in the revision. We selected an ensemble model as in general these models have shown superior performance to single models. We used LASSO regression in part to reduce the number of models in the final ensemble. We found that 13/20 models had a nonzero coefficient, implying that these 13 models were independent predictors of lung-related mortality. We added text to the Methods to clarify this important point:

Methods, page 12, paragraph 2:

“LASSO regression coefficients are given in Supplementary Table 2. We found that 13 out of 20 models had a nonzero LASSO regression coefficient and were included in the final ensemble. A comparison of these 13 single models vs. the ensemble model is given in Supplementary Figure 11”

We also added Supplementary Figure 11 comparing the ensemble model with each individual model. We found that no single individual model outperformed the ensemble model across internal and external testing datasets.

Supplementary Figure 11:

Supplementary Figure 11: Concordance index for individual deep learning models and the ensemble model

Supplementary Figure 11: Concordance index for individual deep learning models and the ensemble model for lung-related mortality in PLCO and NLST testing datasets. The results demonstrate the superior performance of our ensemble approach.

Comment 1.9: How are the 20 models combined? The authors mentioned LASSO but there are insufficient details.

We agree with the reviewer and added additional details about how the LASSO regression model was trained to the Methods.

Methods, page 14, paragraph 2:

“The final model was an ensemble of 20 models combined using a LASSO regression. LASSO regression coefficients were fit using the glmnet package in R, and hyperparameters for the LASSO were selected based on minimum mean-squared error in an internal 10-fold cross validation. LASSO regression coefficients for each of the 20 models are given in Supplementary Table 2 and performance of each of the 20 models in Supplementary Figure 11.”

We also added the LASSO regression coefficients to Supplementary Table 2

Supplementary Table 2: Model architectures and hyperparameters for each of the 20 ensemble models

Model Number	Batch Size	Epochs	Architecture	Learning Rate(s)	Mixup Data Augmentation	LASSO Coefficient
LASSO Intercept						49.85
0	16	58	inceptionv4	0.003938972	0	-1.16
1	32	41	inceptionv4	7.299E-03, 5.546E-04	0	0
2	64	56	inceptionv4	5.474E-03, 2.504E-04	0	2.00
3	128	50	tiny	0.006511004	0	-1.67
4	32	56	tiny	5.230E-03, 2.272E-04	1	1.80
5	64	64	inceptionv4	4.522E-03, 2.682E-04	1	-0.49
6	256	66	inceptionv4	1.759E-03, 3.753E-05	1	0
7	32	53	resnet34	1.353E-03, 5.631E-05	1	0
8	16	40	inceptionv4	0.008946	0	0

9	16	55	resnet34	4.423E-03,9.210E-04	0	-0.22
10	256	51	inceptionv4	7.781E-03,2.395E-04	0	0.98
11	128	43	inceptionv4	7.310E-03,5.340E-04	0	0
12	128	59	resnet34	0.002123776	0	-0.12
13	128	49	inceptionv4	1.920E-03,7.070E-04	1	0
14	64	43	tiny	5.309E-03,1.857E-04	0	2.95
15	32	61	inceptionv4	0.009207081	0	2.79
16	256	46	tiny	0.005282977	0	-0.84
17	32	43	tiny	3.581E-03,9.715E-04	0	2.71
18	64	56	resnet34	2.528E-03,3.341E-04	1	0.98
19	256	48	inceptionv4	5.941E-03,3.818E-04	1	0

Supplementary Table 2: Model architectures and hyperparameters for each of the 20 ensemble models. Models with multiple learning rates were trained using a two-phase approach where the model was trained for 50% of epochs with a high learning rate, then fine-tuned with a lower learning rate. Batch size was randomly chosen from the set [32,64,128,256], epochs were chosen uniformly at random from [40,70], architecture was chosen randomly from the set [tiny, resnet34, inceptionv4], and learning rate was chosen uniformly at random from [1e-3, 1e-6].

Finally, we referenced Supplementary Table 2 in the Methods.

Methods. page 12. paragraph 2:

“The output of these 20 models was combined into a single prediction using a LASSO regression model trained on the hyperparameter tuning dataset. LASSO regression coefficients are given in Supplementary Table 2...”

Comment 1.10: The authors are encouraged to describe their deep learning models clearly in the main text. Such important information shall not be tucked away in the supplement.

We agree with the reviewer that more detailed information about the model should be included in the main text. To address this comment the following changes have been made in the revised manuscript.

Methods, page 13, paragraph 2:

“Instead of a binary target variable of lung-related mortality, we defined age-adjusted labels reflecting the risk of lung disease mortality based on prevalent risk factors for those that did not die of a lung-related disease during follow-up. We posit that these labels are more informative than assigning a “0” for all individuals that did not die, regardless of their underlying risk profile. We define these age-adjusted labels according to the following equation:

$$LR = CA + (E-D)$$

where LR is the Lung-Risk label, CA is the current chronologic age, E is expected age-at-death based on US social security life tables⁴⁶. D corresponds to age-at-death based on A) actual, observed age at death for those that died of a lung disease or lung cancer or B) an individual's predicted age at death due to lung disease/lung cancer based on a survival regression model trained using data from the control (no imaging) arm of the PLCO trial (n=77,444) for those who did not die (Supplementary Table 42). This regression model used prevalent risk factors as input to estimate the age an individual would die of lung-related disease. This model accurately estimated age at death due to lung disease with a concordance index of 0.82 (95% CI [0.817 – 0.825]) for all lung disease mortality and 0.84 (95% CI [0.835 – 0.845]) for lung cancer death. The output of this model was only used to train the deep learning model. This approach was similar to our previous work in which we trained a model to estimate a general biological chest x-ray age¹⁴. Cause of death was adjudicated by the trial based on death certificates and the National Death Index. These Lung-Risk labels were only used for training. The reported results in the testing datasets are all based on the actual observed deaths during follow-up.

The model was trained using a mean-squared error loss function with the ADAM optimizer. The number of epochs for training was selected uniformly at random between 40 and 70. This was independently chosen for each of the 20 models. Training was done using an Ubuntu Linux workstation with an AMD 2960x 24-core CPU with 128 GB of system RAM, and a single NVIDIA RTX A600 GPU with 48 GB of GPU RAM. The model was developed using fastai v2.5.3, PyTorch v1.10, and CUDA v11.2. The full source code and programming environment are freely available at <https://aim.hms.harvard.edu/cxr-lungrisk>

The final model was an ensemble of 20 models combined using a LASSO regression. LASSO regression coefficients were fit using the glmnet package in R, and hyperparameters for the LASSO were selected based on minimum mean-squared error in an internal 10-fold cross validation. LASSO regression coefficients for each of the 20 models are given in Supplementary Table 2 and performance of each of the 20 models in Supplementary Figure 11.

Comment 1.11: The preprocessing of the images is completely missing. How did you convert them from DICOM to PNG? What are the dimensions of the images? Did you use any cropping and/or padding? What data augmentations did you use?

We apologize for leaving this out. We have added image preprocessing, and data augmentation details to Supplementary Methods.

Supplements, Methods, page 2, paragraph 1:

“We used the same image preprocessing pipeline as in our prior work¹. For PLCO and NLST participants, only the baseline radiograph was used, which was acquired upright in posterior-anterior projection. PLCO radiographs were provided as scanned films in .tif file format with protected health information redacted using black pixels. PLCO radiographs were converted to Portable Network Graphics (.png) format using ImageMagick v6.8.9-9. For NLST, all chest radiographs were available in Digital Imaging and Communications in Medicine (DICOM) format. Radiographs for BLCS patients were acquired through clinical care and available in DICOM format from the hospital's Picture Archiving and Communications System. We converted NLST and BLCS DICOM files to .tif using DCMTK v3.6.1 and then to .png using ImageMagick to maintain consistency with the aforementioned PLCO radiographs. To increase sample size in BLCS, both posterior-anterior or anterior-posterior radiographs were included if they were taken up to 3 months prior to histologically confirmed lung cancer diagnosis. For training, images were rescaled to 224 pixels on the short-axis and randomly cropped to 224 x 224 before input into the model. This random cropping was done each time the image was fed to the model as a form of data augmentation. Additional data augmentations used for training included mixup data augmentation, up to 20 degrees of random rotation, up to 20% zoom in/out, and up to 40% brightness/contrast adjustments.”

1. Lu, M. T. et al. Deep Learning to Assess Long-term Mortality From Chest Radiographs. JAMA Netw Open 2, e197416 (2019).

Comment 1.12: On line 132, the authors claim, “CXR Lung-Risk remained independently associated with lung disease mortality in both subpopulations”. But in supplemental Fig. 3B, the lung-risk hazard ratio of the [65, 75] category is not significant in the multivariate model. This sentence is misleading.

We apologize for this confusion and for providing a misleading figure. In fact, the hazard ratio for the 65-70-year old category is significant (HR: 1.69; 95% CI 1.01-2.77). We have updated the figure in the revised manuscript as shown below.

Supplementary Figure 3: Independent testing of the CXR Lung-Risk model in ever smokers and never smokers of PLCO to estimate lung disease mortality

Some minor comments:

Comment 1.13: On line 302, what do you mean by “baseline radiographs”?

We apologize for the imprecise wording and have provided a definition for “baseline radiograph” when first mentioned in the text.

Methods, page 12, paragraph 2:

“..only baseline radiographs, defined as the initial radiograph obtained at the enrollment (T0) exam were used.”

Comment 1.14: The concept of CXR Lung-Risk shall be introduced in the beginning of the article or else it's rather confusing.

Again, we apologize for omitting a more detailed description. We have added more detailed information in the revised manuscript.

Introduction page 3, paragraph 3:

“In this study, we developed a CNN (CXR Lung-Risk) to identify individuals at high risk for lung disease mortality. The only input to the model is a single existing chest radiograph and the output is a risk for lung disease mortality expressed in years meaning, if the model outputs a risk of 75 years this is an equal risk of lung disease mortality as the risk of an average 75-year-old individual. We tested the prognostic value of CXR Lung-Risk in ...”

Reviewer #2 (Remarks to the Author):

This is an interesting manuscript using deep learning applied to routine chest radiographs and showing an ability to predict lung death.

Comment 2.1: I am not a methodology expert but assume the convolutional neural network analyses are done appropriately. I wonder how different the model had looked if one of the other cohorts had served as generator of the model, though.

We thank the reviewer for the positive remarks and considering the impact of an alternative study design. We drew our development dataset from PLCO because it was the largest (N=52,320, compared to 5,414 for NLST and 407 for BLCS). A large number of examples is needed to train these models, so we took the pragmatic approach of using PLCO for development and the other cohorts for independent testing. In addition, PLCO is a cohort of asymptomatic individuals with a risk profile similar to the general population whereas NLST and BLCS participants represent high-risk smokers or persons with diagnosed lung cancer, which were not suitable to develop a model for the general population. To improve clarification of this important point, we have added a statement in the manuscript.

Methods, page 12, paragraph 2:

Model development: The CXR Lung-Risk model was developed using data from the Prostate, Lung, Colorectal, Ovarian (PLCO) Cancer Screening Trial^{23,24}, as it was the largest available dataset in the current study.

I have a number of critical comments.

Comment 2.2: First, I am not sure how "routine" these radiographs are as they are all part of 3 large trials/studies so I assume there is some underlying standardisation.

We thank the reviewer for pointing this out and agree that "routine" can be confusing, as only BLCS radiographs were obtained in clinical routine whereas PLCO and NLST radiographs were acquired in a clinical trial scenario. We therefore deleted the word "routine" throughout the revised manuscript.

Comment 2.3: Secondly, the population in all three cohorts consist of heavy smokers at high risk of lung cancer or subjects who have already been diagnosed/treated for lung cancer. This is a fairly narrowly defined population and I would assume that in many places those at high risk are now offered CT screening and those with recent lung cancer are likely followed up with frequent and/or more advanced imaging?

Again, we thank the reviewer for pointing this out. In fact, PLCO is a cohort of asymptomatic adults with a smoking profile of the general population who are not at increased risk for lung cancer. In contrast, NLST and BLCS participants comprise heavy smokers with an elevated risk profile or prevalent disease and, as indicated by the reviewer, should or will get CT imaging for screening/follow-up imaging. For clarification we have added the following text to the limitations.

Limitations, page 11, paragraph 1:

“In addition many patients at increased risk for lung cancer or prevalent disease get other imaging tests including serial computed tomography. Whether specifically tailored models to estimate prognosis using this imaging data needs to be investigated in additional studies.

Comment 2.4: I am not convinced that the effects can not simply be explained by residual confounding from smoking and other risk factors. As I read methods, the authors adjust for smoking but not pack-years. Pack-years are associated with outcome and changes such as emphysema are also associated with pack-years and like a factor of importance within the model.

We thank the reviewer for noting this point and apologize for the confusion. Our results in both PLCO and NLST datasets are adjusted for pack-years. Only for BLCS, this information was not collected systematically, so we could not evaluate this. This important point was added to the revised manuscript.

Methods, page 15, paragraph 2:

“Unlike PLCO and NLST, pack-years were not available in BLCS for all individuals and not included in the analysis.”

Comment 2.5: Also, did the authors have access to lung function testing. It would be interesting to see if these variables influenced the model or perhaps weakened the predictive value.

We thank the reviewer for this excellent suggestion. These data were available in a subset of BLCS (n=348). We have added correlation analysis and Cox regression analysis including FEV1 in the revised manuscript, showing that CXR Lung-Risk predictions remain significant after correction for FEV1.

Results, page 8, paragraph 3:

“Finally, the relation between CXR Lung-Risk and FEV1 was investigated in BLCS, which showed a moderate negative correlation (Pearson’s $r=0.30$; $p<0.001$; Supplementary Figure 10A). When adding FEV1 to a multivariable model with the same demographic and clinical risk factors as above, the association between CXR Lung-Risk and lung-cancer specific mortality remained significant for the >75 years old category (hazard ratio: 1.92 [1.04-3.55]; $p=0.04$; Supplementary Figure 10B).”

Supplementary Figure 10: Correlation and association of CXR Lung-Risk and FEV1 in BLCS patients

Supplementary Figure 10: A) Correlation between CXR Lung-Risk and FEV1 in a subset of BLCS patients with available lung function tests. B) Forest plots show univariable and multivariable adjusted hazard ratios with 95% confidence intervals for the different CXR Lung-Risk groups. Multivariable models are adjusted for: chronological age, sex, race, obesity, smoking status, cancer stage, treatment and FEV1. BLCS=Boston Lung Cancer Study; FEV1=forced expiratory volume during the first second; HR=hazard ratio

Reviewer #3 (Remarks to the Author):

The paper describes the development and evaluation of a deep learning algorithm to estimate the risk of lung disease mortality using chest radiographs. Overall, the paper is clearly written and the approach seems appropriate.

We would like to thank the reviewer for these positive remarks.

The main questions are whether the improvement in model performance is sufficient to merit broader adoption, which is compounded by questions about the ease of clinical application of the proposed tool.

We agree with the reviewer's opinion. In the current study we provide evidence that the proposed model can improve risk assessment in heavy smokers and in patients with prevalent lung cancer. Whether this translates to other lung-related diseases needs to be investigated in future studies, as mentioned in the limitations. Clinical application could be performed with minimal disruption of current workflows by implementing the model into the EMR or PACS to automatically analyze existing chest x-rays without the need for human input. We have revised the manuscript accordingly.

Limitations, page 11, paragraph 1:

“Whether there is a similar prognostic value for early detection/prognosis of other lung diseases such as COPD or asthma or even broader adoption remains to be seen.”

Discussion, page 10, paragraph 2:

“Implementing a tool like CXR Lung-Risk into the EMR or PACS to automatically extract this currently unused information could help to increase the diagnostic value of this imaging test.”

Comment 3.1: Population of interest. The model is developed using the PLCO cancer screening trial data on asymptomatic individuals age 55-74 enrolled at 10 US sites between 1993-2001. The model is validated in a hold-out sample from this population and two other, higher-risk data sets, which is appropriate. One question: was the hold-out sample from PLCO selected chronologically (the newest data) or randomly? Chronological selection is preferred as it better matches clinical reality (we develop model now but apply it to new patients).

We thank the reviewer for this comment. The model was tested in two completely independent datasets and demonstrated good performance. The internal testing dataset in PLCO was selected randomly and not based on chronological dates as we consider this the best and most unbiased approach to test generalizability of the model. In addition, given the study period from 1993-2001. The chest x-rays from the two independent testing cohorts were obtained later between 2002-2004 for NLST and up to 2016 for BLCS patients. To clarify we added the following information to the Methods.

Methods, page 14, paragraph 3:

“The first testing dataset comprised a random sample of 10,155 asymptomatic individuals from PLCO aged 55-74 (20% of individuals; median follow-up=17.0 [IQR 14.8-19.0] years) not used for model development. “

Comment 3.2: Outcome of interest. Outcome is defined as lung disease mortality. Has this outcome been captured consistently across the evaluation samples?

We thank the reviewer for highlighting this important point. For PLCO and NLST, outcomes were assessed via annual questionnaires, communication with next of kin and the National Death Index. Cause of death was determined using ICD-9 codes. For BLCS patients, death was verified by trained study staff via manual chart review and only collected for lung cancer mortality but not other lung disease. For clarification, we have added this important information to the results in addition to the method section.

Results, page 5, paragraph 1:

“PLCO was a multicenter randomized controlled trial of chest radiography for cancer screening in asymptomatic individuals aged 55-74 years enrolled at 10 US sites from 1993 through 2001. Outcomes were assessed via annual questionnaires, communication with next of kin and the National Death Index. Cause of death was determined using ICD-9 codes. II) Participants from the NLST25 chest radiograph arm (n=5,414; median follow-up=11.9 [IQR 7.3-12.3] years). NLST was a randomized controlled trial that enrolled heavy smokers (≥ 30 pack years) aged 55-74 years for lung cancer screening via chest CT vs. chest radiograph at 21 US sites from 2002 through 2004. Similar to PLCO, outcomes were assessed via annual questionnaires, communication with next of kin and the National Death Index and ICD 9 codes were used to determine cause of death. III) patients from the BLCS (n=407; median follow-up=3.4 [IQR 1.5-7.2] years), which is an ongoing multicenter observational epidemiologic cohort registry of patients with histologically confirmed lung cancer. Mortality was verified by study staff via manual chart review and was available for lung cancer-specific mortality only. An overview of the study design and analyses is provided in Figure 1.”

Comment 3.3: Time horizon. The 3 studies in question have differing follow-ups with medians ranging between 17, 12 and 3 years. It is not clear from the descriptions in the paper what was the time horizon for predictions. The authors express the risk score in years, having made a conversion from % risk to age of a person with a given level of risk. But level of risk needs a specified time horizon.

We apologize that we have omitted such important information. The output of the model is 18-year lung disease mortality risk, the maximum available in the PLCO development dataset. We have added this information in the revised manuscript.

Methods, page 12, paragraph 2:

“...the output is an estimated risk of 18-year lung disease mortality...”

Comments 3.4 and 3.5: Predictors. The primary predictor is the CXR Lung-Risk built based on convolutional neural networks (CNN). It was added to the model containing standard risk factors. Valid approach.

Mathematical model. CNN for CXR Lung-Risk and Cox PH model for adding CXR Lung-Risk to other risk factors. This is a solid approach

We would like to thank the reviewer for these positive remarks.

Comment 3.6: Model performance. While CXR Lung-Risk expressed as age appears to be very strong in terms of hazard ratios, we need to realize that this form of presentation combines the

effect of the images with that of age. As the authors explain, a risk of 75 means the risk of a person who is 75 years old. This means the average of all 75-year-olds and those with risk at the level of average 75-year-old. When compared to the risk of 65 which is defined similarly, this combines age and image information. This explains why the very large hazard ratios translate into small increments in the C-indices and leave us with the question to what extent it is worth the effort.

We thank the reviewer for this comment. In fact, the only input to the model is the chest radiograph image. No additional information e.g. chronological age is input into the model. For clarification, we have added this information to the introduction and legend of the overview figure (Figure 1) in addition to the description in the methods and discussion. Please also refer to comment 1.14.

Introduction page 3, paragraph 3:

“In this study, we developed a CNN (CXR Lung-Risk) to identify individuals at high risk for lung disease mortality. The only input to the model is a single existing chest radiograph and the output is a risk for lung disease mortality expressed in years meaning, if the model outputs a risk of 75 years this is an equal risk of lung disease mortality as the risk of an average 75-year-old individual. We tested the prognostic value of CXR Lung-Risk in ...”

Figure 1: Overview of the study design. A) The CXR Lung-Risk model was developed in PLCO. The only input to the model is the raw chest radiograph image; the model output is an estimated risk of lung disease mortality. Independent testing was performed in a held-out subset of PLCO participants, individuals enrolled in NLST and patients with histologically confirmed lung cancer from the BLCS. B) The prognostic performance of the CXR Lung-Risk model was evaluated and compared to clinical risk factors in all datasets.

Discussion, page 9, paragraph 1:

“In this study, we propose a deep learning convolutional neural network that estimates the risk of lung disease mortality from a chest radiograph image as the only input.”

Methods, page 12, paragraph 2:

“The only input to the proposed CXR Lung-Risk model is a chest radiograph image; the output is an estimated risk of 18-year lung disease mortality....”

Comment 3.7: Clinical Decision Support. The authors leave us with the hazard ratios and C-indices but do not propose how their model could be used in practice. Should we look at risk reclassification?

We thank the reviewer highlighting this point. In the revised manuscript we have added more detailed and concrete examples, where the use of the proposed model could be helpful to optimize clinical workflows.

Discussion, page 9, paragraph 2:

“Furthermore, the substitution of chronological age with the biological age captured by CXR-Lung-Risk in existing risk calculators/predictors may provide more accurate decision support. For example, lung cancer prediction models (either for selection of screening candidates or cancer risk prediction of solitary pulmonary nodules) commonly include chronological age³², and substitution with CXR-Lung-Risk may improve the utility and accuracy of these clinical risk predictions and help for risk reclassification beyond current methods.”

32. Tammemägi, M. C. et al. Selection criteria for lung-cancer screening. *N. Engl. J. Med.* 368, 728–736 (2013).

Discussion, page 10, paragraph 2:

“For example, only around 5% of eligible Americans are screened for lung cancer³³⁻³⁵. Here, the proposed model could help to flag individuals at high risk to prompt risk discussion and encourage entry into screening programs. Furthermore, it has been reported that approximately 70% of individuals with COPD are underdiagnosed³⁶ with the risk for increased morbidity and mortality. In this context, CXR Lung-Risk could be deployed to automatically notify the treating physician to schedule a follow-up visit for the patient to investigate possible causes and discuss potential interventions, such as a full pulmonary function test and the use of bronchodilator. As no human input is necessary, CXR Lung-Risk could be used with minimal disruption of current clinical workflows and automatically analyze the latest radiograph of a patient at high speed and low additional cost³⁷. As such, CXR Lung-Risk could serve as an early warning system to triage patients into existing screening and chronic pulmonary disease pathways, and to both provide more accurate risk assessments for those programs and increase adherence to guideline-based therapies.”

33. Wang, G. X. et al. Barriers to Lung Cancer Screening Engagement from the Patient and Provider Perspective. *Radiology* 290, 278–287 (2019).

34. Jemal, A. & Fedewa, S. A. Lung Cancer Screening With Low-Dose Computed Tomography in the United States-2010 to 2015. *JAMA Oncol* 3, 1278–1281 (2017).

35. Richards, T. B. et al. Lung Cancer Screening Inconsistent With U.S. Preventive Services Task Force Recommendations. *Am. J. Prev. Med.* 56, 66–73 (2019).

36. Diab, N. et al. Underdiagnosis and Overdiagnosis of Chronic Obstructive Pulmonary Disease. *Am. J. Respir. Crit. Care Med.* 198, 1130–1139 (2018).

Comment 3.8: How are these patients treated today? What can be done for them differently if we had a risk score, one with imaging data?

Please refer to the response of the previous comment 3.7.

Reviewer #4 (Remarks to the Author):

In this paper, the authors developed a deep learning model, named CXR Lung-Risk, that predicts the risk of lung disease mortality from a routine CXR image. The deep learning model, which is actually an ensemble of several AI models, has been trained on 80% of the data from the PLCO trial, and is validated on the remaining 20% of the PLCO trial (n=10155), a dataset from the NLST dataset (heavy smokers) (n=5414), and a clinical multi-center dataset from the Boston Lung Cancer Study (n=407). The authors found that the model showed a graded association with lung disease mortality, and this association remained present after adjusting for known risk factors, including chronological age, smoking, and radiologic findings.

The final claim of this manuscript is that the results demonstrate that deep learning can identify individuals at high risk of lung disease mortality from easily obtainable, low-cost, and routine CXR images.

The manuscript is well written, clear, and newsworthy. The paper is related to a previous publication in which the lung age was predicted from a routine chest radiography image using machine learning. The model is externally validated on two datasets (NLST and BLCS). The PLCO and NLST are datasets which are both quite old and therefore, the additional validation on the data from the BLCS study is an important addition. The manuscript contains no evidence about the clinical impact of the CXR Lung-Risk AI model, but this is the topic of future studies.

We would like to thank the reviewer for these positive remarks.

I have the following comments and points of criticism:

Comment 4.1: Please add in what years the CXRs from the BCLS study were obtained in the Methods section.

We agree with the reviewer and apologize for omitting this important information. The CXRs of the BLCS patients were acquired between 2004-2016. We have added the missing information in the method section.

Methods, page 14, paragraph 5:

“For this study, only patients with early stage (I-III) and a diagnosis between 2004-2016 were included.”

Comments 4.2: There are too little details regarding the training and architecture of the models. In my opinion, more details should be added. What architectures were used specifically? What loss function was used? On what hardware was this trained? etc

We apologize for leaving out this critical information. In the revised manuscript we provide more detailed information on the model architecture, loss function and hardware as outlined below.

Methods page 13, paragraph 3:

“The model was trained using a mean-squared error loss function with the ADAM optimizer. The number of epochs for training was selected uniformly at random between 40 and 70. This was independently chosen for each of the 20 models. Training was done using an Ubuntu Linux workstation with an AMD 3960x 24-core CPU with 128 GB of system RAM, and a single NVIDIA RTX A6000 GPU with 48 GB of GPU RAM. The model was developed using fastai v2.5.3, PyTorch

v1.10, and CUDA v11.2. The full source code and programming environment are freely available at <https://aim.hms.harvard.edu/cxr-lungrisk>”

and Methods, page 12, paragraph 2:

“..., CXR Lung-Risk was built as an ensemble model to reduce variance in the output.^{38,39} The ensemble consisted of 20 CNNs. The model architecture for each CNN was chosen randomly from a set of architectures popular in medical image analysis (inceptionv4, resnet34, tiny⁴⁰⁻⁴²). Hyperparameters for each model were randomly selected during training (Supplementary Table 2), as random hyperparameters have been shown to improve performance in ensemble learning by reducing correlation between models.⁴³”

43. Wenzel, F., Snoek, J., Tran, D., & Jenatton, R. Hyperparameter Ensembles for Robustness and Uncertainty Quantification. *Adv. Neural Inf. Process. Syst.* 33, (2020).

Comment 4.3: The manuscript is inconsistent in the terms multivariable vs multivariate. The methods section mentions multivariable while the rest of the manuscript uses multivariate. What is correct? Please check carefully and adjust.

We apologize for this inconsistent wording. The correct term is multivariable, which is now used consistently throughout the revised manuscript.

Comment 4.4: This claim in the Discussion is too strong and without evidence: "If CXR-Lung risk, a personalized risk estimate, rather than chronological age was used for determining the type of treatment, these patients would have significantly more likely received a tumor-directed therapy." Please adjust.

We thank the reviewer for pointing this out. We have rephrased the sentence as indicated below.

Discussion, page 9 paragraph 2:

“If CXR-Lung risk, a personalized risk estimate, rather than chronological age was used, this could affect treatment decisions for tumor-directed therapy.”

Comment 4.5: What about the quality of the chest radiographs? The model is trained with radiographs acquired between 1993 and 2001. Do the authors expect problems with data drift when this model is applied in more modern datasets? I think this needs discussion.

Again, we thank the reviewer for pointing this out. Indeed, the training dataset is rather old. This was necessary to accrue long term mortality followup (up to 18 years). However, independent testing in BLCS with CXRs acquired up to 2016 shows robust performance of the model. In addition, validating the model in even more recent datasets is difficult due to necessary long-term follow-up. To address this comment we have added a statement in the limitations.

Limitations, page 11, paragraph 1:

“Finally, PLCO chest radiographs were collected from 1993-2001 and available as scanned films. Whether this has an impact on model accuracy in more modern datasets was not systematically analyzed in the current study. However, independent testing in BLCS, where the most recent radiographs were acquired in 2016, showed robust performance.”

Comment 4.6: Please add subgroup analysis for different age groups on PLCO and NLST as well in the manuscript, as was done for the BLCS study.

We apologize that this information has not been added before. The missing analyses were incorporated in the revised version of the manuscript.

Results page 6 paragraph 4:

“In addition, stratified analyses by sex and chronological age (<65 years old vs. ≥65 years old) are provided in the Supplements (Supplementary Figures 4 and 5), which revealed similar results for all investigated subgroups.”

Supplementary Figure 5: Independent testing of the CXR Lung-Risk model in PLCO stratified by chronological age to estimate lung disease mortality

Supplementary Figure 5: Kaplan-Meier survival analysis shows a graded association between CXR Lung-Risk groups and lung disease mortality in A) chronologically younger (<65 years old) and B) chronologically younger (≥65 years old). Forest plots show univariable and multivariable adjusted hazard ratios with 95% confidence intervals for the different CXR Lung-Risk groups. Multivariable models are adjusted for: chronological age, sex, race, smoking status, pack years, body mass index, prevalent diabetes mellitus, hypertension, history of stroke, myocardial infarction, cancer and 9 radiologic findings as described in the methods.

*p=*** (p<0.001); PLCO=Prostate, Lung, Colorectal, Ovarian Cancer Screening Trial; HR=hazard ratio*

and Methods page 7, paragraph 2:

“As for PLCO, similar results were seen in NLST participants for sex and age stratified analyses (Supplementary Figures 7 and 8) and lung cancer-specific mortality for the entire cohort (Supplementary Figure 6B).”

Supplementary Figure 8: Independent testing of the CXR Lung-Risk model in NLST stratified by chronological age to estimate lung disease mortality

Supplementary Figure 8: Kaplan-Meier survival analysis shows a graded association between CXR Lung-Risk groups and lung disease mortality in A) chronologically younger (<65 years old) and B) chronologically older (≥65 years old). Forest plots show univariable and multivariable adjusted hazard ratios with 95% confidence intervals for the different CXR Lung-Risk groups. Multivariable models are adjusted for: chronological age, sex, race, smoking status, pack years, body mass index, prevalent diabetes mellitus, hypertension, history of stroke, myocardial infarction, cancer and 9 radiologic findings as described in the methods. Atelectasis and bone/chest wall lesions (for chronologically younger only) and lung opacity and lymphadenopathy (for chronologically older only) were removed from the model because coefficients converged due to too few events. $p=***$ ($p<0.001$); NLST=National Lung Screening Trial; HR=hazard ratio

REVIEWER COMMENTS

Reviewer #1 (Remarks to the Author):

The authors have addressed all my comments.

Reviewer #2 (Remarks to the Author):

Thank you for your responses to my queries.

Reviewer #3 (Remarks to the Author):

The authors answered several of my questions. A handful of concerns remain:

1. I am still not able to reconcile the large hazard ratios and the small increments in the c indices. It might be related to a number of things (size of age categories, use of age as a time scale etc.). Also, the effects drop substantially in the 2nd validation cohort - is this related to the duration of follow-up? I would like the authors to explain all these issues.
2. Differences in c indices should be presented with a 95% CI and no statistical test should be given because the one available is known to be incorrect for nested models.
3. It would be nice to see a reclassification analysis on the data you have available to see what the clinical impact of your approach could be.

Reviewer #4 (Remarks to the Author):

I would like to thank the authors for their rebuttal. It addressed my comments and concerns adequately, and I think the manuscript is ready for publication.

Response to the Reviewers' Comments

We thank the reviewers for their detailed feedback and have responded in blue. The revised manuscript was updated to incorporate these important new insights and data. For more details, please find our point-by-point response below.

Reviewer #1 (Remarks to the Author):

The authors have addressed all my comments.

We thank the reviewer for this positive assessment.

Reviewer #2 (Remarks to the Author):

Thank you for your responses to my queries.

You are welcome.

Reviewer #3 (Remarks to the Author):

The authors answered several of my questions. A handful of concerns remain:

Comment 3.1: I am still not able to reconcile the large hazard ratios and the small increments in the c indices. It might be related to a number of things (size of age categories, use of age as a time scale etc.). Also, the effects drop substantially in the 2nd validation cohort - is this related to the duration of follow-up? I would like the authors to explain all these issues.

We thank the reviewer for highlighting this important missing information and apologize that this was not incorporated in the previous revision. As the reviewer points out, the discrepancy between the large hazard ratios and the relatively small improvement in c indices is related to several aspects:

1) the number of individuals in the high risk group (>75-years) with the high hazard ratios is much smaller than the middle (≥ 65 - <75-years) and low risk groups (<65-years). See first lines of Table 1 below:

Table 1: Patient demographics and clinical risk factor in PLCO and NLST participants

Variables	PLCO testing dataset				NLST			
	Entire dataset	Lung-Risk <65 years	Lung-Risk ≥ 65 -<75 years	Lung-Risk >75 years	Entire dataset	Lung-Risk <65 years	Lung-Risk ≥ 65 -<75 years	Lung-Risk >75 years
N	10,155	7182	2667	306	5414	2600	2433	381

To address this point we have added the following to the limitations of the manuscript.

Limitations, page 11, paragraph 1:

"In PLCO and NLST there is a discrepancy between the relatively small increase in the c indices in nested model comparison and the large hazard ratios, especially in the high risk groups (CXR

Lung-Risk >75-years), which is likely explained by the significantly different number of individuals in the different risk groups.”

2) The risk profile of individuals in PLCO and NLST. While PLCO enrolled asymptomatic individuals from the general population, NLST only included lung cancer screening eligible individuals with a smoking history of ≥ 30 pack years.

3) Median length of follow-up is significantly different between the two cohorts (17.0 [14.8-19.0] years vs. 11.9 [7.6-12.3] years), as pointed out by the reviewer.

To reconcile these differences, we modified Supplementary Figure 6 by limiting the PLCO cohort to lung cancer screening eligible individuals (≥ 30 pack years) instead of any individual with a smoking history. We also used a 12-year follow-up cutoff to match NLST. Revised Kaplan-Meier plots and hazard ratios (reproduced below) are much more comparable between both cohorts.

Results, page 6, paragraph 5:

“In a sensitivity analysis, we tested the association between CXR Lung-Risk and lung cancer-specific mortality in current or former smokers (quit <15 years ago) with a smoking history of ≥ 30 pack years to allow for a better comparison to individuals enrolled in NLST (see below). CXR Lung-Risk showed a graded and independent association with lung cancer-specific mortality after adjusting for demographics and clinical risk factors (Supplementary Figure 6B).”

Supplementary Figure 6: Independent testing of the CXR Lung-Risk model in the PLCO testing dataset limited to current or former smokers (quit <15 years ago) with a smoking history of ≥ 30 pack years and in NLST(all current or former smokers who quit <15 years ago with a smoking history ≥ 30 pack years) to estimate lung cancer-specific mortality.

Comment 3.2: Differences in c indices should be presented with a 95% CI and no statistical test should be given because the one available is known to be incorrect for nested models.

We agree with the reviewer and have revised the manuscript accordingly.

Results, page 6, paragraph 2:

“To test whether CXR Lung-Risk adds incremental value to a baseline multivariable model with the same covariates but without CXR Lung-Risk, nested Cox proportional hazard models were compared. Adding CXR Lung-Risk to the baseline model resulted in a modest improvement to estimate lung disease mortality compared to the baseline model alone (c-index: 0.83 [95% CI 0.81-0.85] vs. 0.81 [95% CI 0.79-0.83]).”

Results, page 7, paragraph 1:

“In addition, CXR Lung-Risk showed a modest improvement in estimating lung disease mortality when added to the multivariable model of demographics and clinical risk factors alone (c-index: 0.76 [95% CI 0.74-0.78] vs. 0.72 [95% CI 0.70-0.74]).”

Results, page 7, paragraph 3:

“Likewise to the other testing data sets, a small improvement to estimate lung cancer-specific mortality was found for CXR Lung-Risk when comparing multivariable nested Cox models with and without CXR Lung-Risk (c-index: 0.76 [95% CI 0.72-0.80] vs. 0.75 [95% CI 0.71-0.79]).”

Comment 3.3: It would be nice to see a reclassification analysis on the data you have available to see what the clinical impact of your approach could be.

We thank the reviewer for this excellent suggestion and have added a reclassification analysis in the revised manuscript.

Results, page 8, paragraph 1:

“To investigate the potential clinical impact of CXR Lung-Risk in patients with lung cancer we calculated risk reclassification tables based on the CXR Lung-Risk categories and chronological age (<65-years old vs. ≥65-years old) (Table 3). We found increasing mortality rates by CXR Lung-Risk categories in both those <65-years of chronologic age and ≥65-years.”

Table 3: Risk reclassification of lung cancer-specific mortality based on risk categories defined by chronological age and CXR Lung-Risk

		CXR Lung-Risk		
		<65-years	≥65-years - <75-years	≥75-years
Chronological Age	<65-years	24 (33.8%)	42 (40.8%)	9 (45%)
	≥65-years	5 (27.8%)	54 (42.2%)	40 (59.7%)

Reviewer #4 (Remarks to the Author):

I would like to thank the authors for their rebuttal. It addressed my comments and concerns adequately, and I think the manuscript is ready for publication.

We would like to thank the reviewer for this opinion.

REVIEWERS' COMMENTS

Reviewer #3 (Remarks to the Author):

Thank you for your responses to my outstanding comments.